# Revisiting the role of Dcc in visual system development with a novel eye clearing method

**Robin J Vigouroux, Quénol Cesar, Alain Chédotal\*,
Kim Tuyen Nguyen-Ba-Charvet\***

Institut de la Vision, Sorbonne Université, INSERM, CNRS, Paris, France

**Abstract** The Deleted in Colorectal Carcinoma (Dcc) receptor plays a critical role in optic nerve development. Whilst Dcc is expressed postnatally in the eye, its function remains unknown as *Dcc* knockouts die at birth. To circumvent this drawback, we generated an eye-specific *Dcc* mutant. To study the organization of the retina and visual projections in these mice, we also established EyeDISCO, a novel tissue clearing protocol that removes melanin allowing 3D imaging of whole eyes and visual pathways. We show that in the absence of *Dcc*, some ganglion cell axons stalled at the optic disc, whereas others perforated the retina, separating photoreceptors from the retinal pigment epithelium. A subset of visual axons entered the CNS, but these projections are perturbed. Moreover, *Dcc*-deficient retinas displayed a massive postnatal loss of retinal ganglion cells and a large fraction of photoreceptors. Thus, Dcc is essential for the development and maintenance of the retina.

**\*For correspondence:**
alain.chedotal@inserm.fr (AC);
kim.charvet@inserm.fr (KTN-B-C)

**Competing interests:** The authors declare that no competing interests exist.

## Introduction

Deleted in Colorectal Carcinoma (Dcc) is a transmembrane receptor discovered as a potential tumor suppressor (*Pierceall et al., 1994*). Dcc binds the extracellular matrix protein Netrin-1 (*Keino-Masu et al., 1996*; *Serafini et al., 1994*). When bound to Netrin-1, Dcc activates downstream signaling partners such as MAP kinase, focal adhesion kinase (FAK) or Src kinases which ultimately influence cytoskeleton dynamics and cell motility (*Mehlen et al., 2011*; *Ren et al., 2004*). However, Dcc is also a dependence receptor (*Mehlen et al., 2011*; *Mehlen et al., 1998*) which triggers cell death in absence of Netrin-1 (*Llambi et al., 2005*; *Mehlen et al., 2011*). Although direct evidence linking Dcc to tumorigenesis was obtained in mice (*Castets et al., 2012*; *Krimpenfort et al., 2012*), mutations in *DCC* were also identified in patients suffering from rare neurodevelopmental disorders including congenital mirror movements (*Depienne et al., 2011*; *Srour et al., 2010*) and corpus callosum dysgenesis (*Jamuar et al., 2017*; *Marsh et al., 2017*).

Dcc is present in the developing central nervous system (CNS) and controls axon guidance and cell migration in the spinal cord, as well as in multiple brain areas (*Belle et al., 2014*; *Fazeli et al., 1997*; *Fothergill et al., 2014*; *Laumonnerie et al., 2014*; *Schmidt et al., 2014*; *Srivatsa et al., 2014*; *Yee et al., 1999*). Dcc expression persists in the postnatal and adult CNS where it is involved in synaptogenesis (*Horn et al., 2013*; *Manitt et al., 2013*) and myelination (*Jarjour et al., 2008*). In vivo evidence supporting Dcc function in CNS development was primarily obtained using *Dcc* knockout (KO) mice (a null allele) which lack Dcc in all cells (*Fazeli et al., 1997*). One of the most striking phenotype, a hypoplasia of the optic nerve, was reported in the visual system (*Deiner et al., 1997*; *Shi et al., 2010*). In *Dcc* KO embryos, a large fraction of the retinal ganglion cell (RGC) axons which connects the eye to the brain via the optic nerve is unable to exit the retina (*Deiner et al., 1997*). The presence of its ligand Netrin-1 at the optic disc, the exit point of the retina, together with the presence of hypoplasic optic nerves in *Ntn1* hypomorph mutant embryos suggested that Netrin-1

acts as a long range cue attracting Dcc-expressing RGC axons towards the optic nerve head (*Deiner et al., 1997*; *Shi et al., 2010*).

Dcc expression persists in RGC axons after they exit the retina and even postnatally (*Shi et al., 2010*) but as *Dcc* null mice die at birth (*Fazeli et al., 1997*), its function at later stages of visual system development is unknown. In *Xenopus laevis* tadpoles, Netrin-1 attracts RGC axons toward the optic disc (*Shewan et al., 2002*) and promotes RGC axon arborization and synapse formation within the tectum (*Manitt et al., 2009*). Importantly, recent studies using conditional knockout strategies have revisited the role of Netrin-1 in axon guidance at the CNS midline and suggested that Netrin-1 does not act as a long-range attractive cue for axons (*Dominici et al., 2017*; *Moreno-Bravo et al., 2019*; *Varadarajan et al., 2017*; *Wu et al., 2019*). Although the conditional ablation of *Dcc* in specific neuronal classes recapitulates the axon guidance defects found in *Dcc* KO embryos in some systems (*da Silva et al., 2018*; *Peng et al., 2018*; *Zelina et al., 2014*), there is also evidence for a non-cell autonomous role in cortical projection neurons (*Welniarz et al., 2017*). Here, we have generated eye-specific *Dcc* mutants which are fully viable. We found that these mutant mice display severe optic nerve hypoplasia as well as axon pathfinding defects in visual centers that persist postnatally. These defects are accompanied by a massive elimination of RGCs and the death of a large subset of photoreceptors. We also describe EyeDISCO, a novel tissue-clearing method allowing whole-eye immunostaining that could open novel perspectives in vision research.

## Results

### Dcc is broadly expressed in the developing retina

The presence of Dcc in RGCs had been previously shown (*Deiner et al., 1997*; *Shi et al., 2010*). Here, we first studied the temporal and spatial expression pattern of Dcc in the developing mouse retina using an antibody recognizing the C-terminal region of Dcc (*Mazelin et al., 2004*). The first Dcc-immunoreactive cells were detected at embryonic day 11 (E11) in the dorsal half of the retina and co-localized with the early RGC marker, Islet1 (*Figure 1A–B*; n = 3)(*Austin et al., 1995*). The specificity of the antibody was supported by the absence of staining in retinas from *Dcc* KO embryos (*Figure 1—figure supplement 1A–D*; n = 3). To get a better understanding of the spatial distribution of Dcc-expressing cells, we carried out whole-mount immunohistochemistry of Dcc and Islet1 at E11 followed by iDISCO+ clearing (*Renier et al., 2016*) and confocal microscopy for three-dimensional (3D) rendering. This confirmed that the early expression of Dcc at E11 is restricted to the medio-dorsal retina (*Figure 1C–D*; n = 4). By E12, Dcc expression expanded and still co-localized with Islet1-positive cells (*Figure 1E–H*; n = 3) but not with the transcription factor Sox2 (*Figure 1—figure supplement 1E–G*; n = 3), a retinal progenitor cell (RPC) marker (*Kamachi et al., 1998*). Thus, Dcc is only present in post-mitotic cells in the early retina. By E15, Dcc expression was broader and spanned both the apical and basal retina. Dcc-positive cells were either immunoreactive for Islet1 (*Figure 1I–L*; n = 3), or for Cone-rod homeobox protein Crx (Crx; *Figure 1—figure supplement 1H–J*; n = 3), which is expressed by post-mitotic photoreceptor cells.

### A novel eye-specific Dcc mutant

Eyelid opening and retinal maturation in the mouse occur around the second postnatal week. As *Dcc* full knockout mice (*Dcc*$^{-/-}$) die a few hours following birth, whether Dcc could be involved in later stages of visual system development is unknown. To address this issue, we used mice driving the Cre-recombinase under the promoter of *Dickkopf-3* (*Dkk3*), a gene specifically expressed by RPCs (*Sato et al., 2007*). Crossing these *Cre* driver mice with mice carrying a *tdTomato* reporter, we found that the *Cre* was successfully driven in the retina (and not in other parts of the CNS) as early as E9 (n = 3) and with a complete retinal recombination by E10 (a time which precedes the onset of Dcc expression) (*Figure 2A–C*; n = 3). To ensure that Dcc protein was completely removed in the *Dkk3:cre;Dcc*$^{fl/fl}$ mice prior to optic nerve exit, we carried out a Dcc immunolabeling on E11 retinas. Dcc$^+$ cells were found in *Dcc*$^{fl/fl}$ retinas, but not in *Dkk3:cre;Dcc*$^{fl/fl}$ retina (*Figure 2—figure supplement 1A–D*; n = 3 for each genotype).

We next questioned whether the retinal progenitor pool (expressing Ceh-10 homeodomain-containing homolog, Chx10, also known as visual system homeobox2, Vsx2) (*Liu et al., 1994*) as well as the generation of RGCs (Islet1) were maintained in *Dcc* mutants. At E12, no obvious defects were

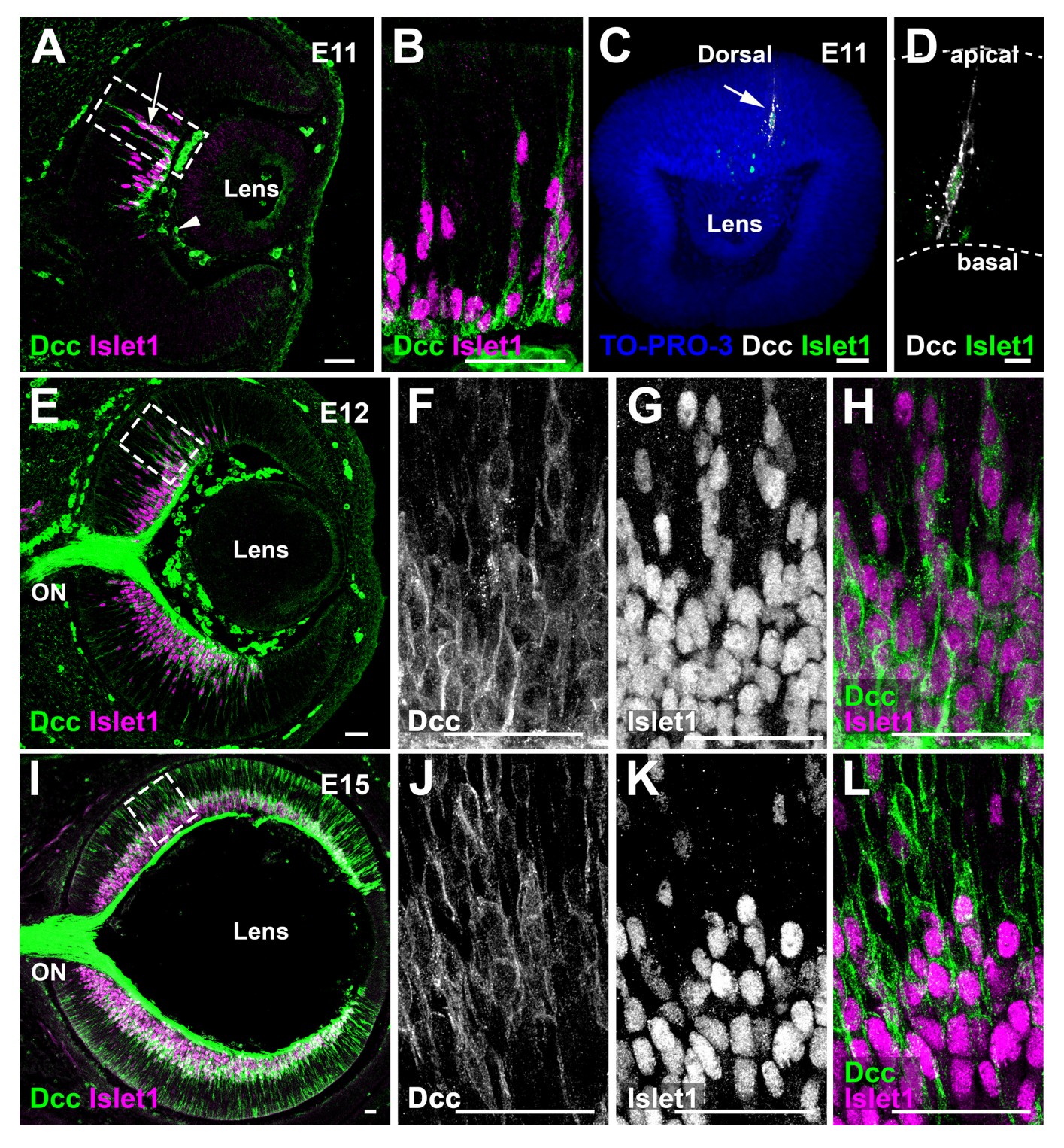

**Figure 1.** Dcc is broadly expressed in the mouse retina. (**A**) Immunohistochemistry (IHC) of Dcc (arrow, green) and the retinal ganglion cell marker Islet1 (magenta), at E11. Non-specific Dcc signal is seen in mesenchymal cells (arrowhead) (**B**) is a high magnification image. (**C,D**) Whole-mount IHC of an E11 eye for Dcc (white, arrow) and Islet1 (green) counterstained with the nuclear marker TO-PRO-3 (blue). (**E**) Dcc (green) and Islet1 (magenta) IHC at E12. (**F–H**) are high magnification images. (**I–L**) Dcc (green) and Islet1 (magenta) IHC at E15. Scale bars: (**A–H**) 50 µm, (**I,K**) 30 µm, (**J,L**) 10 µm. ON, Optic Nerve.

The online version of this article includes the following figure supplement(s) for figure 1:

*Figure 1 continued on next page*

*Figure 1 continued*

**Figure supplement 1.** Early Dcc expression is restricted to post-mitotic cells of the retina.

seen in mutants (n = 3) compared to *Dcc^fl/fl* littermates (*Figure 2D–O*; n = 3). Furthermore, EdU incorporation showed no proliferation rate defects in progenitor cells (*Figure 2—figure supplement 1E–L*, *Figure 2—source data 1*; n = 3).

## Complexity of RGC guidance defects in Dcc KO revealed by a novel eye clearing method

Abnormal RGC projections towards the optic nerve head were previously reported in *Dcc^-/-* embryos (*Deiner et al., 1997*) however, the long-term consequence of these defects on the retina are unknown and the exact spectrum of RGC projection defects have not been precisely studied. This is primarily due to the difficulty of inferring complex axon trajectories from simple retinal sections or retinal flat-mounts. In recent years, several tissue clearing protocols have been implemented to study the 3D cellular organization of complex organs such as the brain (*Klingberg et al., 2017*; *Renier et al., 2016*; *Richardson and Lichtman, 2015*; *Tainaka et al., 2018*; *Tomer et al., 2014*; *Vigouroux et al., 2017*). However, these methods do not remove melanin from the layer of retinal pigment epithelium (RPE) cells which cover the retina (as early as E12), thereby blocking light. Therefore, successful eye clearing remains a burning issue in the field (*Susaki and Ueda, 2016*). The current solution to this problem is to dissect out the RPE but this does not maintain eye integrity. To solve this problem, we devised a novel tissue clearing protocol (see methods), EyeDISCO, that completely clears embryonic and adult mouse eyes (*Figure 3A–B'*).

Pioneer RGC axons, born at E11, extend their projections ventrally along the choroid fissure and into the presumptive optic disc (*Goldberg, 1977*; *Silver, 1984*). To investigate whether *Dcc* deletion induced a defect in pioneer axon pathfinding, we carried out whole-mount immunostaining of Tag1 (Transient axonal glycoprotein 1, also known as Contactin 2) on E12 embryos followed by EyeDISCO clearing (*Figure 3A–B*). Tag1 is expressed by all sensory axons including RGC axons (*Chatzopoulou et al., 2008*). In *Dcc^fl/fl* mice, Tag1^+ RGC projections extended ventrally towards the optic disc and into the optic nerve where they fasciculated (*Figure 3C–E*; n = 3). However, in *Dcc* cKO mice some RGC projections extended ventrally to the optic disc and stalled, forming thick RGC axon bundles (*Figure 3F–H*; n = 4). In addition, several RGCs misprojected and extended dorsally through the retina and into the sub-retinal space. Therefore, eye-specific deletion of *Dcc* leads to pioneer RGC projection defects.

To further assess RGC projections in *Dkk3:cre;Dcc^fl/fl* mutant embryos, we carried out Tag1 immunostaining on whole E16 embryonic heads followed by EyeDISCO clearing (*Figure 4A–C*). Using manual segmentation (see methods) of the visual pathways (retina, optic nerve, optic chiasm, and optic tracts), we specifically isolated these structures from the rest of the head (*Figure 4D*). We next questioned whether RGC axon guidance defects in *Dcc^-/-* embryos were phenocopied in *Dkk3:cre; Dcc^fl/fl* embryos. In both *Dcc^-/-* (n = 8 nerves) and *Dkk3:cre;Dcc^fl/fl* mice (n = 6 nerves) there was a significant reduction in optic nerve volume compared to Wildtype mice (*Figure 4E–H*, *Figure 4—source data 1*; n = 8 nerves). Heterozygous deletion of *Dcc* (*Dkk3:cre;Dcc^lox/+*) had no effect on optic nerve volume (n = 4 nerves).

We then investigated whether the optic nerve (ON) hypoplasia observed in *Dcc* mutant mice was a result of fewer RGCs projecting into the optic nerve head. Analysis of E16 *Dkk3:cre;Dcc^fl/fl* mutant retinas immunolabeled for Tag1 showed that RGC projections exhibited multiple patterns. A subset of RGCs projected basally along the retinal lamina and into the optic nerve (*Figure 4G*; *Figure 4—video 1*). Another subset extended apically into the sub-retinal space at many sites all along the eye (*Figure 4G*; *Figure 4—video 1*). Interestingly, these projections separated the future photoreceptor cell outer segments from contacting with the RPE which is critical for their survival (*Figure 4I–L*; *Strauss, 2005*). Analysis of retinal cryosections at P7 showed that aberrant RGC axons, expressing ßIII-tubulin, separated the photoreceptor outer segments from the RPE (*Figure 4—figure supplement 1A–D*). Of note, these abnormal projections either stalled in the sub-retinal space or joined the optic nerve. Importantly, manual segmentation showed that although these projections exited at

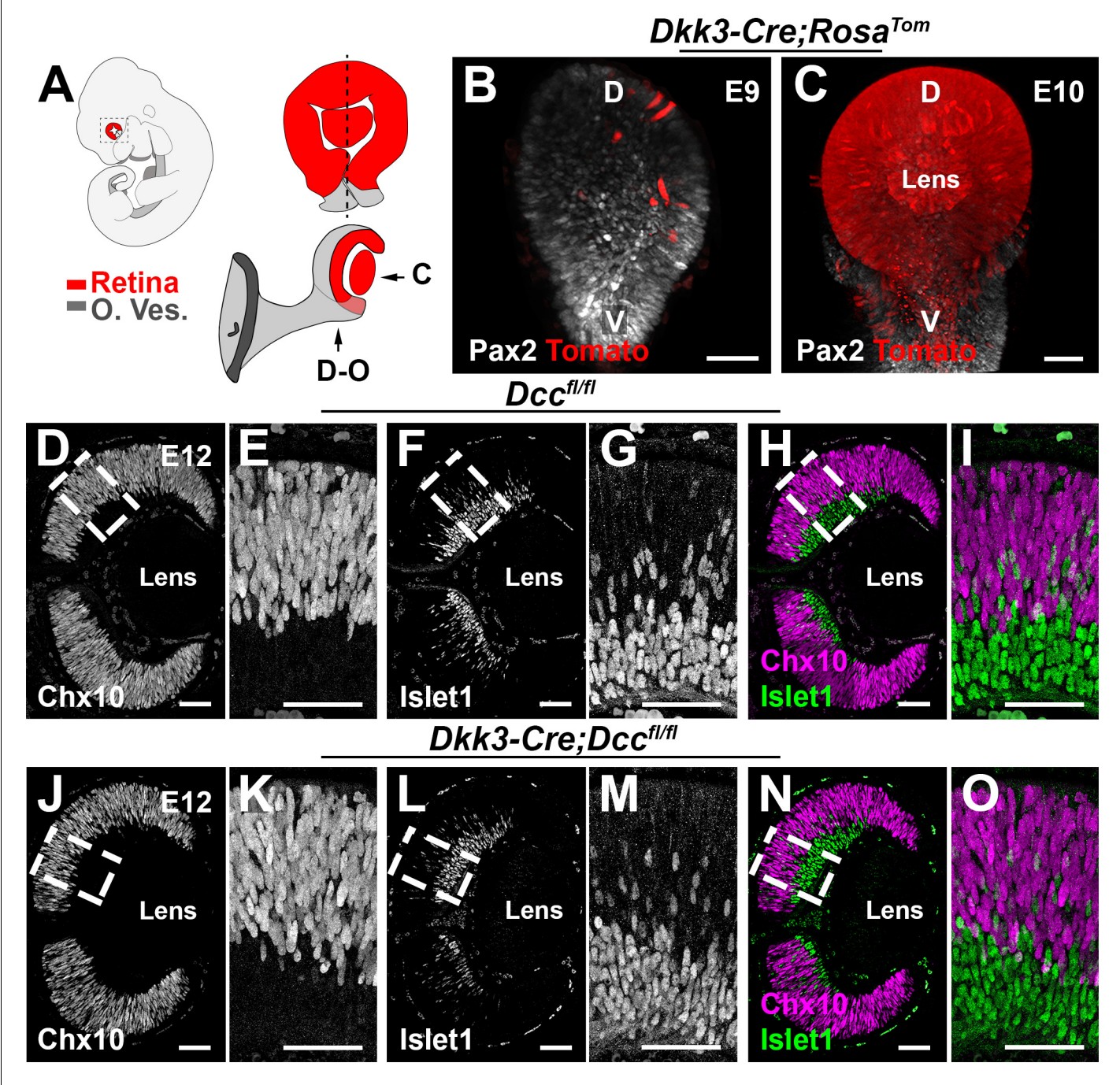

**Figure 2.** Retina-specific inactivation of Dcc. (A) Schematic representation of an E10 embryo (upper left) showing the neural retina (red) and the optic vesicle (gray). (B,C) Lateral views of the eye from *Dkk3:cre;Rosa^Tom* E9 and E10 embryos after whole-mount labeling for dsRed (Rosa tomato) and Pax2 (optic vesicle). (C) Represents the whole-mount lateral visualization, whereas (D–O) shows the orientation of the sagittal cryosections. (D–O) Cryosections of E12 *Dcc^fl/fl* and *Dkk3:cre;Dcc^fl/fl* embryos labeled for Chx10 (magenta) and Islet1 (green). Scale bars: (B,C) 30 μm, (D–O) 50 μm. D, dorsal; V, ventral; O. Ves, optic vesicle.

The online version of this article includes the following source data and figure supplement(s) for figure 2:

**Source data 1.** Eye-specific loss of Dcc does not impact the proliferation of retinal progenitor cells.

**Figure supplement 1.** *Dcc* deletion does not impact early retinal proliferation.

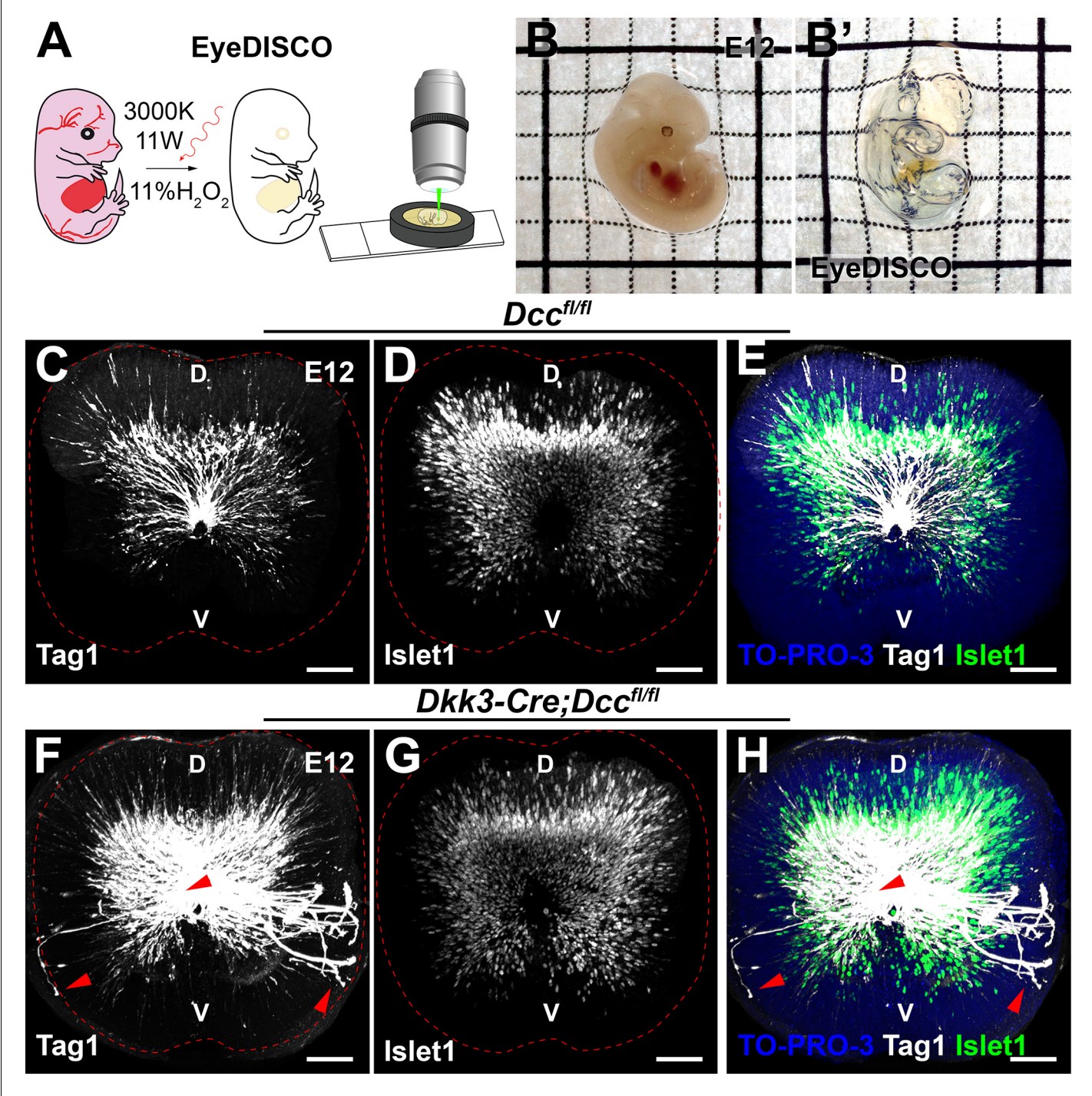

**Figure 3.** EyeDISCO, a novel tissue clearing protocol for embryonic eye visualization. (**A**) Schematic representation of the EyeDISCO protocol. The embryo is dehydrated in methanol and incubated in an 11% $H_2O_2$ solution irradiated with a 3000 °K warm white light. The sample is then included in a homemade chamber for confocal microscopy and 3D rendering. (**B,B'**) Images of an E12 embryo before and after EyeDISCO clearing. (**C–H**) Lateral view of whole-mount E12 eyes immunolabeled for the RGC axon marker, Tag1 (white) and the RGC nuclear marker (Islet1) counterstained with the nuclear marker TO-PRO-3 (blue). Several axons misproject in the *Dkk3:cre;Dcc^fl/fl* mutants (red arrowheads) or are stalled at the optic disc. Scale bars: (**C–H**) 50 μm. D, dorsal ; V, ventral.

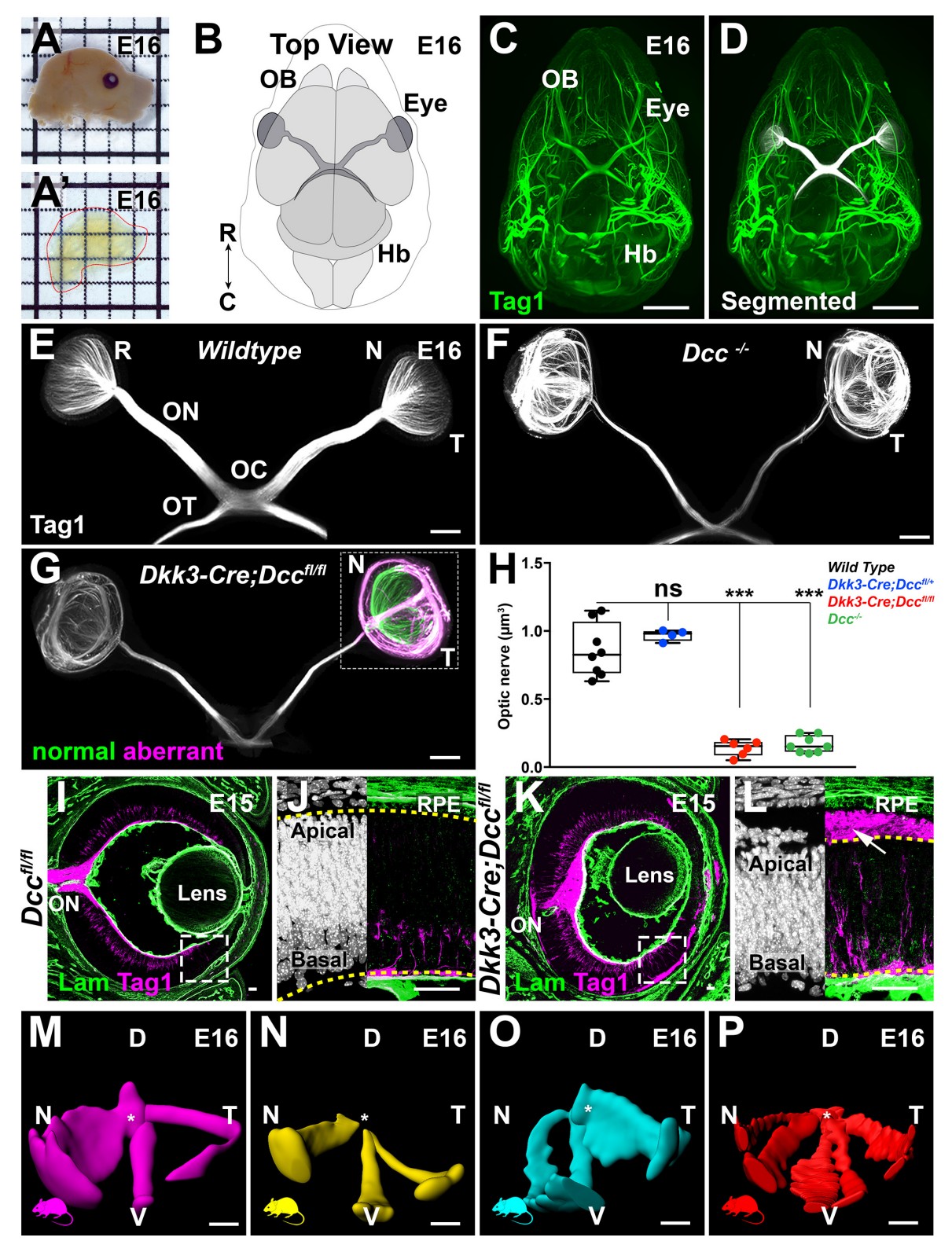

**Figure 4.** Early Dcc deletion leads to major intra-retinal axonal misprojections. (A–A') Illustrates the clearing efficiency of the EyeDISCO protocol of an E16 mouse head. (B) Schematic representation of a top view of an E16 head, OB: olfactory bulb, Hb: hindbrain, R: rostral, C: caudal. (C–G) IHC whole-mount of E16 heads labeled for Tag1 (green). (D) E16 embryo head labeled with Tag1 in (C) with the visual projections manually segmented (white). (E–G) Segmented visual projections in E16 Wildtype, *Dcc⁻/⁻*, and *Dkk3:cre;Dcc^fl/fl* embryos. R: retina, ON: optic nerve, OC: optic chiasm, OT: optic tract, N:
*Figure 4 continued on next page*

Figure 4 continued

nasal, T: temporal. (H) Quantification of optic nerve volumes (μm³) represented as a box plot, whiskers represent min to max values. *Dcc⁻/⁻* (0.165 ± 0.022 μm³; n = 8 nerves) and *Dkk3:cre;Dcc^fl/fl* embryos (0.140 ± 0.23 μm³; n = 6 nerves) displayed a significant reduction in optic nerve volume compared to Wildtype embryos (0.858 ± 0.07 μm³; n = 8 nerves) (p=0.0002 and p=0.0007 respectively, Mann-Whitney test). Heterozygous deletion of *Dcc* (*Dkk3:cre;Dcc^lox/+*) had no effect on optic nerve volume (0.97 ± 0.02 μm³; p=0.2828; n = 4 nerves, Mann-Whitney test). Results were considered non-significant (ns) when p>0.05. ***=p < 0.001. (I,J,K,L) Cryosections of E15 *Dcc^fl/fl* and *Dkk3:cre;Dcc^fl/fl* eyes immunolabeled for Laminin (green) and Tag1 (magenta). (I) In controls, RGC projections (magenta) grow circumferentially and enter the optic nerve. (K) In *Dkk3:cre;Dcc^fl/fl* embryos, RGC projections perforate the retina and stall at the optic disc, some projections manage to exit into the ON. (J,L) High magnification images. (L) In *Dkk3:cre;Dcc^fl/fl* embryos, RGC projections extend apically and invade the sub-retinal space (arrow), separating the RPE and the apical retina (future photoreceptor outer segments). (M–P) Individual masks of aberrantly projecting RGCs across different mutants. Asterisks show the optic nerve. D, Dorsal; V, Ventral; N, Nasal; T, Temporal. Scale bars: (C,D) 1000 μm, (E–G) 300 μm, (I–P) 150 μm.

The online version of this article includes the following video, source data, and figure supplement(s) for figure 4:

**Source data 1.** Retina-specific deletion of Dcc leads to a significant reduction in optic nerve volume.
**Source data 2.** Early Netrin-1 deletion in the retina leads to a significant reduction in optic nerve volume.
**Figure supplement 1.** Eye-specific deletion of Dcc and Netrin-1 leads to major retinal defects.
**Figure 4—video 1.** Embryonic RGC projections are perturbed in *Dcc cKO* mice.
https://elifesciences.org/articles/51275#fig4video1

multiple sites, they fasciculated and formed major bundles that spanned the medial and ventral retina (*Figure 4M–P*; n = 4).

To determine whether the RGC axon phenotype observed in *Dcc* cKO mice was dependent on Netrin-1 signaling, we generated an eye-specific conditional deletion of *Netrin-1* (*Dkk3;cre;Ntn1^fl/fl*). At E16, we observed a significant reduction in optic nerve volume in *Ntn1* null mice (*Ntn1⁻/⁻*) (*Moreno-Bravo et al., 2018*) (n = 9; *Figure 4—figure supplement 1F,H*; *Figure 4—source data 2*). Likewise, E16, *Dkk3;cre;Ntn1^fl/fl* embryos displayed a major reduction in optic nerve volume (n = 5) (*Figure 4—figure supplement 1G,H*; *Figure 4—source data 2*).

Altogether, eye-specific deletion of *Dcc* prompted early RGCs to misproject apically through the retina. Furthermore, eye-specific deletion of Netrin-1 phenocopied the defect observed in *Dcc* cKO mice.

## Retinal projections in the brain are altered in eye-specific Dcc mutants

The observation that a significant proportion of RGCs were still able to project their axons into the optic nerve in the absence of Dcc prompted us to study their projections within the brain. Mouse RGCs connect to at least 40 different brain nuclei (*Morin and Studholme, 2014*). To get the most comprehensive and faithful image of visual projections we used axonal tracers and iDISCO+ whole-brain clearing (*Renier et al., 2016*). Mice were injected intravitreally with AlexaFluor-555 or Alexa-Fluor-647-conjugated cholera toxin β-subunit (CTB) (see methods) allowing to distinguish ipsi- and contra-laterally projecting RGCs. iDISCO+ cleared brains were imaged using light sheet fluorescence microscopy (LSFM) (*Figure 5A,B*, *Figure 5—video 1*).

We first focused on the primary visual system which consists of the optic nerve, optic chiasm, optic tract, lateral geniculate nuclei, and the superior colliculus. In order to quantify differences, we carried out automatic segmentation of CTB-stained nuclei using Imaris software (see materials and methods). From this segmentation we generated surfaces with Imaris that retraced RGC projections in each visual system nuclei (*Figure 5C*). Volume of each surfaces were extracted in μm³ and quantified to analyze main differences. Analysis of *Dkk3:cre;Dcc^fl/fl* mutant brains showed that despite some heterogeneity between the amount of reduction between eyes, optic nerve volumes were dramatically reduced by about half at P15 and 1 month (*Figure 5D,E,F*; *Figure 5—source data 1*; n = 4 for *Dcc* cKO; n = 5 for *Dcc^fl/fl*). In addition, optic tract volumes were significantly reduced at P15 (n = 5) and 1 month (*Figure 5G*; *Figure 5—source data 1*; n = 5 for *Dcc* cKO; n = 7 for *Dcc^fl/fl*). Thus, early RGC projection defects and RGC death translate to a major reduction in fiber volume projecting into the visual nuclei postnatally.

During RGC projections refinement, which occurs between P5 and P15 in mice, ipsilaterally projecting RGCs segregate to form a robustly stereotypic L-shaped rostral patch in the superior colliculus (*Godement et al., 1984*). We found that at 1 month, the ipsilateral patch represented ~5% (n = 5) of each superior colliculus in *Dcc^fl/fl* mice. Interestingly, there was a significant expansion of

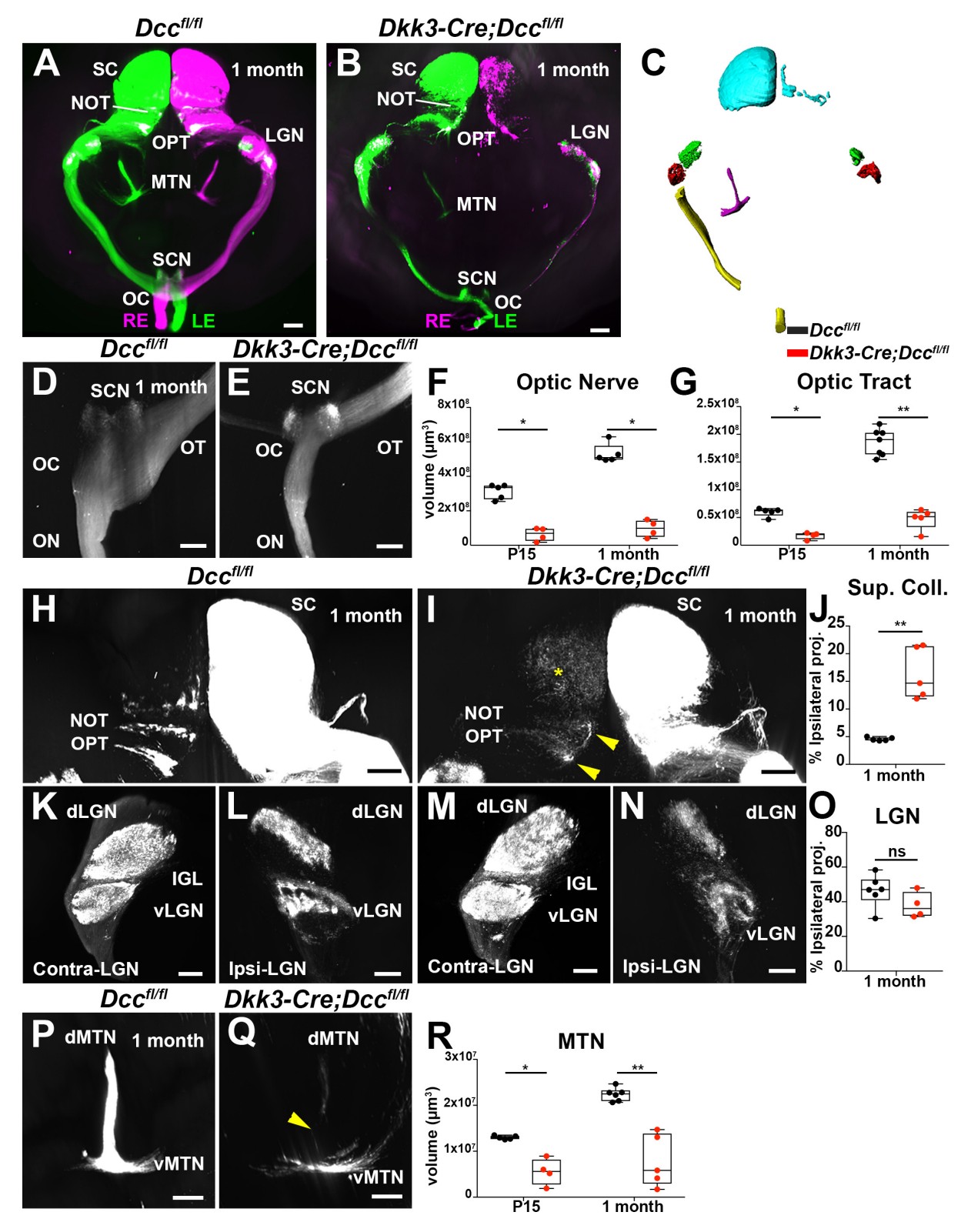

**Figure 5.** Retinal projections in the brain are altered in eye-specific *Dcc* mutants. (**A,B**) Frontal view of 3D-rendered brains using after anterograde axon tracing of visual projections with AlexaFluor-555 and/or AlexaFluor-647-conjugated cholera toxin β subunit (CTB). RE: right eye, LE: left eye, OC: optic chiasm, SCN: supra chiasmatic nucleus, MTN: medial terminal nucleus, LGN: lateral geniculate nucleus, NOT: nucleus of the optic tract, OPT: olivary pretectal nucleus, SC: superior colliculus. (**A**) Control brain, and (**B**) a *Dkk3:cre;Dcc^{fl/fl}* brain. (**C**) Surfaces of manual and automatic segmentation of visual
*Figure 5 continued on next page*

*Figure 5 continued*

projections using Imaris software in a control brain. Superior colliculus (cyan), dorsal lateral geniculate nucleus (green), ventral lateral geniculate nucleus (red), medial terminal nucleus (magenta), optic nerve and optic tract (yellow). (D,E) Top view of the optic nerve, optic chiasm, and optic tract. (F) Quantification of optic nerve volumes at P15 ($6.36 \times 10^6 \pm 1.92 \times 10^6$ µm³; n = 4; compared to $3.12 \times 10^7 \pm 1.92 \times 10^6$ µm³; n = 5; p=0.0159) and 1 month ($9.67 \times 10^6 \pm 2.48 \times 10^6$ µm³; n = 4; compared to $5.32 \times 10^7 \pm 2.51 \times 10^6$ µm³; n = 5; p=0.0159). A Mann-Whitney test was used to measure the significance. (G) Quantification of optic tract volumes (µm³) at P15 ($1.73 \times 10^7 \pm 3.13 \times 10^6$ µm³; n = 4; compared to $5.97 \times 10^7 \pm 3.42 \times 10^6$ µm³; n = 5; p=0,0159) and 1 month ($4.76 \times 10^7 \pm 8.37 \times 10^6$ µm³; n = 5; compared to $1.86 \times 10^8 \pm 9.11 \times 10^6$ µm³; n = 7; p=0.0025). A Mann-Whitney test was used to measure the significance. (H,I) Top view of the superior colliculus. *Dcc* cKO mice display a defasciculated NOT and OPT (yellow arrowhead), as well as an aberrant segregation of ipsilateral RGC projections (yellow asterisk). (J) Percentage of ipsilateral projections normalized to the contralateral projections (volume, µm³) of the superior colliculus. At 1 month, ipsilateral superior colliculus volume represented 4.565 ± 0.1424% (n = 5) in *Dcc*^{fl/fl} mice compared to 16.39 ± 2.091% in *Dkk3:cre;Dcc*^{fl/fl} mice (p=0.0079, n = 5). A Mann-Whitney test was used to measure the significance. (K–N) Frontal view of the contra- and ipsi-lateral geniculate nucleus. dLGN: dorsal lateral geniculate nucleus, vLGN: ventral lateral geniculate nucleus, IGL: inner geniculate leaflet. (F,K) Ipsilateral lateral geniculate nucleus. (O) Percentage of ipsilateral projections normalized to the contralateral projections (volume, µm³) of the lateral geniculate nucleus were not altered in *Dcc* cKO mice (37.89 ± 3.751%; n = 4) compared to control littermates (46.36 ± 3.795%; n = 6; p=0.352) A Mann-Whitney test was used to measure the significance. (P,Q) Frontal view of the medial terminal nucleus, dMTN: dorsal medial terminal nucleus, vMTN: ventral medial terminal nucleus. *Dcc* cKO mice show disturbed projections between the dMTN and the vMTN (yellow arrowhead). (R) Quantification of medial terminal nucleus volumes (ventral and dorsal, µm³). At P15, *Dcc* cKO mice display a reduction ($5.51 \times 10^6 \pm 1.44 \times 10^6$ µm³; n = 4) compared to control ($1.23 \times 10^7 \pm 1.79 \times 10^5$ µm³; n = 5; p=0.0159). At 1 month, this loss was maintained ($7.89 \times 10^6 \pm 2.55 \times 10^6$ µm³; n = 5; compared to $2.24 \times 10^7 \pm 5.95 \times 10^5$ µm³; n = 6; p=0.0043). A Mann-Whitney test was used to measure the significance. Whiskers represent min to max values. *=p < 0.05, **=p < 0.01. Scale bars: (A,B) 1000 µm, (H,I) 500 µm, (D,E,K,L,M,N, P,Q) 300 µm.

The online version of this article includes the following video and source data for figure 5:

**Source data 1.** *Dcc* cKO mice show a significant reduction of RGC projection volumes in multiple brain visual nuclei.

**Figure 5—video 1.** Adult RGC projections display major defects in eye-specific *Dcc* deletion.

https://elifesciences.org/articles/51275#fig5video1

---

this ipsilateral territory to ~16% in *Dkk3:cre;Dcc*^{fl/fl} mice (*Figure 5H,I,J*; *Figure 5—source data 1*; n = 5). On the other hand, analysis of projections within the lateral geniculate nucleus (ventral and dorsal) did not show an expansion in ipsilateral territory (*Figure 5K–O*; *Figure 5—source data 1*).

To date, very little is known about the development of the Accessory Optic System (AOS) (*Osterhout et al., 2015*; *Sun et al., 2015*). We therefore wondered whether deletion of *Dcc* in RGCs could also lead to AOS defects. The most striking effect in *Dcc* cKO mice was observed in the medial terminal nucleus (MTN), which can be subdivided into a ventral (vMTN) and dorsal (dMTN) nucleus (*Lilley et al., 2019*). In *Dcc* mutant mice, RGCs targeted appropriately the vMTN and some RGCs also projected to the dMTN. Projections connecting the vMTN to dMTN were completely absent (*Figure 5P,Q*). In P15 *Dcc* cKO mice, MTN volume was significantly reduced (n = 4) compared to *Dcc*^{fl/fl} mice (*Figure 5R*; *Figure 5—source data 1*; n = 5). This effect was maintained at 1 month in *Dcc* cKO mice (n = 5) compared to control littermates (*Figure 5R*; *Figure 5—source data 1*; n = 6). *Dkk3:cre;Dcc*^{fl/fl} mice also displayed defects in other AOS nuclei such as the nucleus of the optic tract (NOT) and the olivary pretectal tract (OPT). Their projections appeared defasciculated (*Figure 5H,I*). Taken together, eye-specific *Dcc* deletion perturbs RGC axon targeting in the main and accessory visual systems.

## Dcc intracellular signaling is required for retinal projection targeting in the brain

Dcc has been shown to act as a co-receptor to other guidance receptors (*Corset et al., 2000*; *Hong et al., 1999*; *Ly et al., 2008*) suggesting that the observed defect might not directly, or solely, involve Netrin-1/Dcc signaling. To address this question, we studied the visual system of *Dcc*^{Kanga} mice, which bear a mutation in the exon encoding for the intracellular P3 domain of Dcc (*Finger et al., 2002*). This domain is critical for Dcc signaling downstream of Netrin-1 (*Zhang et al., 2018*). As previously described, we were unable to obtain viable *Dcc*^{kanga/kanga} mice (*Welniarz et al., 2017*). Thus, heterozygous Dcc kanga mutants (*Dcc*^{kanga/+}) were crossed with *Dcc* knockout animals (*Dcc*^{+/-}) to generate *Dcc*^{kanga/-} mutants that possess one *Dcc* allele with the Kanga mutation and one null allele.

*Dcc*^{kanga} mice were injected with the CTB tracer and adult brains were cleared and imaged using LSFM for 3D rendering (*Figure 6*; *Figure 6—video 1*). In *Dcc*kanga^{kanga/-} mice (n = 9), the optic nerve volume was significantly reduced compared to *Dcc*^{kanga/+} mice (*Figure 6A–C*; *Figure 6—*

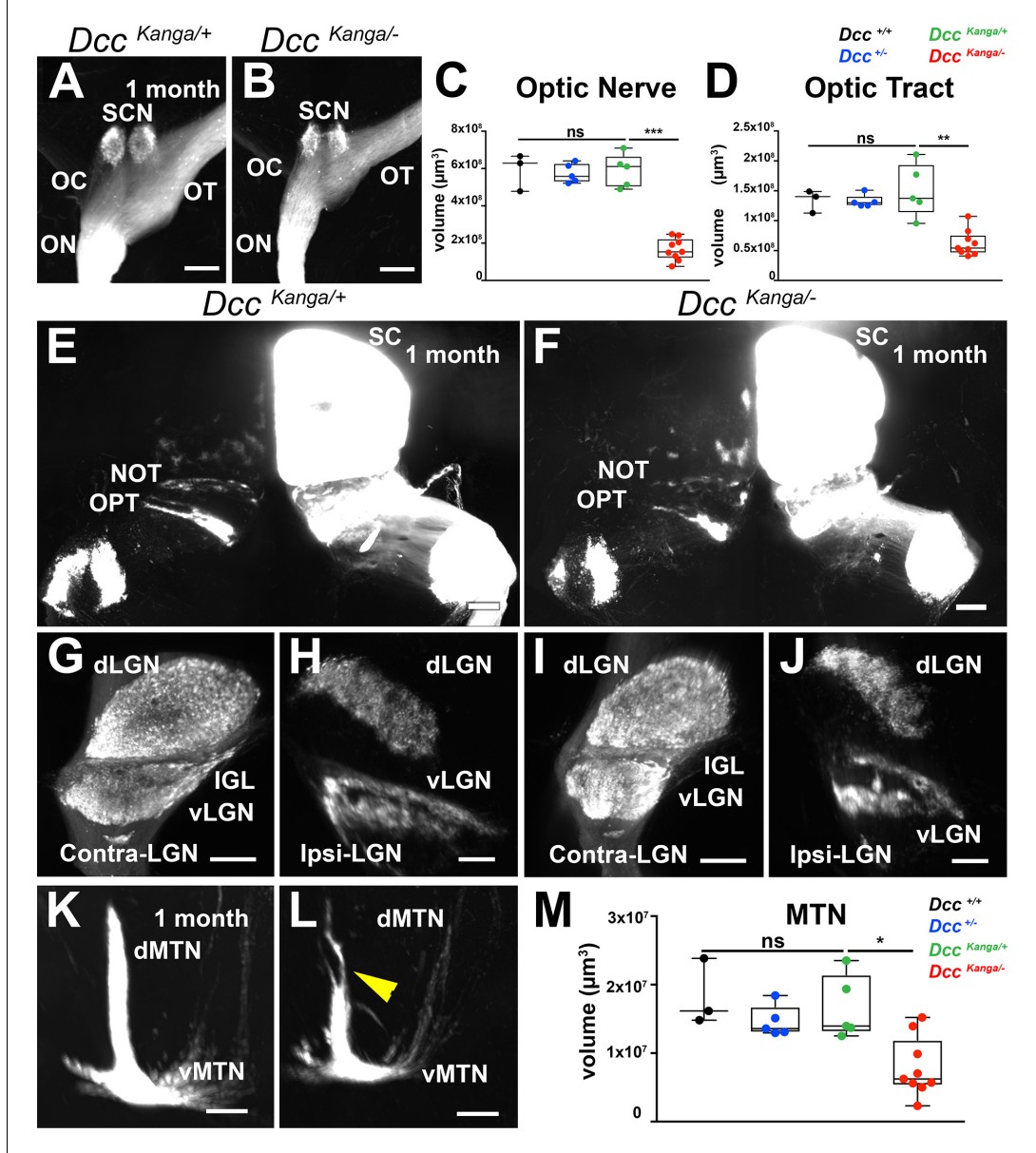

**Figure 6.** *Dcc* signaling is required for RGC projection targeting. (**A–M**) Whole-brain 3D rendering of RGC projections traced using AlexaFluor555-CTB or AlexaFluor647-CTB. (**A,B**) Top view of the optic nerve, optic chiasm, and optic tract of 1 month-old *Dcc*^kanga/+^ and *Dcc*^kanga/-^ mice. OC, optic chiasm; SCN, suprachiasmatic nucleus; OT, optic tract. (**C**) Quantification of optic nerve volume ($\mu m^3$) in *Dcckanga*^kanga/-^ mice ($1.67 \times 10^7 \pm 1.97 \times 10^6$ $\mu m^3$; n = 9) were significantly reduced compared to *Dcc*^kanga/+^ mice ($5.89 \times 10^7 \pm 3.98 \times 10^6$ $\mu m^3$; n = 5; p=0.0010, Mann-Whitney test). (**D**) Optic tract volume ($\mu m^3$) of *Dcckanga*^kanga/-^ mice were also significantly reduced ($0.624 \times 10^8 \pm 7.09 \times 10^6$ $\mu m^3$; n = 9) compared to *Dcc*^kanga/+^ mice ($1.50 \times 10^8 \pm 1.99 \times 10^6$ $\mu m^3$; n = 5; p=0.0020, Mann-Whitney test). (**E, F**) Top view of the superior colliculus. NOT, nucleus of the optic tract; OPT, olivary pretectal nucleus; SC, superior colliculus. (**G,H,I,J**) Frontal view of the contra-lateral LGN. dLGN, dorsal lateral geniculate nucleus; vLGN, ventral lateral geniculate nucleus; IGL, inner geniculate leaflet. (**K,L**) Frontal view of the MTN. dMTN, dorsal medial terminal nucleus; vMTN, ventral medial terminal nucleus. (**M**) Quantification of MTN volume ($\mu m^3$) of *Dcckanga*^kanga/-^ mice ($0.788 \times 10^7 \pm 1.43 \times 10^6$ $\mu m^3$; n = 9) compared to *Dcc*^kanga/+^ littermate controls ($1.66 \times 10^7 \pm 2.09 \times 10^6$ $\mu m^3$; n = 5; p=0.0190, Mann-Whitney test). Results were considered non-significant (ns) if p>0.05. *=p < 0.05; **=p < 0.01; ***=p < 0.001. Scale bars: (**A,B**) 150 μm (**E,F**) 300 μm, (**G,H,I,J, K, L**) 200 μm.

The online version of this article includes the following video, source data, and figure supplement(s) for figure 6:

**Source data 1.** Dcc Kanga mice display a major reduction of RGC projections in multiple brain visual nuclei.

**Source data 2.** Dcc kanga mice show a similar reduction in visual nuclei volumes compared to *Dcc* cKO mice.

**Figure supplement 1.** The reduction in retinal projections observed in *Dcc* cKO mice is phenocopied in *Dcc*^kanga^ mutant mice.

**Figure 6—video 1.** Dcc signaling is required for RGC projections to the MTN.

https://elifesciences.org/articles/51275#fig6video1

source data 1; n = 5). The optic tract volume was also significantly reduced (n = 9) in *Dcckanga*<sup>kanga/-</sup> mice compared to *Dcc*<sup>kanga/+</sup> mice (*Figure 6D*; *Figure 6—source data 1*; n = 5). Of note, no significant differences in optic nerve or optic tract volumes were observed between *Dcc*<sup>+/+</sup>, *Dcc*<sup>+/-</sup>, and *Dcc*<sup>kanga/+</sup> (*Figure 6C,D*; *Figure 6—source data 1*). Interestingly, comparisons between Dcc<sup>Kanga</sup> mice and *Dcc* cKO mice showed no significant differences in optic nerve and optic tract volume (*Figure 6—figure supplement 1A,B*; *Figure 6—source data 2*). We next assessed whether projections within the superior colliculus were disturbed in *Dcckanga*<sup>kanga/-</sup> mice. Unlike in *Dkk3:cre;Dcc*<sup>fl/fl</sup> mice, no major defect was observed in the segregation of ipsilateral projections in *Dcckanga*<sup>kanga/-</sup> mutants compared to *Dcc*<sup>kanga/+</sup> control mice (*Figure 6E,F*). Furthermore, no major defects were observed in both thalamic nuclei of the ventral and dorsal lateral geniculate nucleus (*Figure 6G–J*).

The AOS was also affected in *Dcckanga*<sup>kanga/-</sup> mice, as shown by a significant reduction of the volume of the MTN (n = 9) compared to *Dcc*<sup>kanga/+</sup> littermate controls (*Figure 6K–M*; *Figure 6—source data 1*; n = 5). This reduction in MTN volume was comparable to that observed in *Dcc* cKO mice (*Figure 6—figure supplement 1C*; *Figure 6—source data 2*). RGC projections to the MTN in *Dcckanga*<sup>kanga/-</sup> mutants displayed multiple projection defects (*Figure 6—figure supplement 1D,E*). No defects were observed in other AOS nuclei.

## Eye-specific deletion of Dcc alters retinal layer thickness

To determine whether Dcc could play a role beyond optic nerve formation, we first immunostained postnatal retinas to observe whether Dcc protein was still present. At P0 (n = 3) and P7 (n = 3), Dcc protein was absent from the cell bodies but was still heavily expressed in the postnatal retina (*Figure 7A–F*). It was localized to the neuropil layers of the retina, the inner and outer plexiform layers (*Figure 7A–F*).

Since early RGCs misproject in *Dcc* cKO mice we questioned whether RGCs were affected in postnatal and adult mice. P0, P15, and 1 month-old retinas were flat-mounted and labeled for a pan-RGC marker (*Kwong et al., 2010*), RNA-binding protein with multiple splicing (Rbpms; *Figure 7—figure supplement 1A–F*). Whole-eye immunostaining with Rbpms confirmed the gradual loss of RGCs from P15 to 1 month (*Figure 7—figure supplement 1G–J*; *Figure 7—source data 1*). Of note, RGC loss was homogeneously distributed in mutant retinas. Strikingly, the number of RGCs was dramatically reduced by ~60% in *Dkk3:cre;Dcc*<sup>fl/fl</sup> mice at P0 (n = 4 retinas) when compared to *Dcc*<sup>fl/fl</sup> mice (*Figure 7—figure supplement 1K*; *Figure 7—source data 1*; n = 6 retinas). The number of Rbpms-positive cells decreased to ~82% in P15, *Dkk3:cre;Dcc*<sup>fl/fl</sup> mice (n = 4 retinas) and even further to ~91% at 1 month (*Figure 7—figure supplement 1K*; *Figure 7—source data 1*; n = 6 retinas). Displaced RGCs were not included in our analysis. Thus, eye-specific loss of *Dcc* leads to a dramatic degeneration of RGCs.

Unlike *Dcc*<sup>-/-</sup> mice, *Dkk3:cre;Dcc*<sup>fl/fl</sup> conditional mutants are viable allowing us to investigate the role of Dcc in retinal lamination, which is established postnatally. The inner plexiform layer (IPL) is subdivided into 5 specific lamina named stratum 1 to 5 (S1-S5, S5 being closest to the RGC layer) (*Wässle, 2004*). Since *Dcc* deletion led to early retinal defects, we wondered whether the earliest born amacrine cells, starburst amacrine cells (SACs) (*Voinescu et al., 2009*) displayed defects. In control mice, both ON- and OFF- SACs express choline O-acetyltransferase (Chat) and their dendrites stratify in S2 and S4 of the IPL (*Figure 7I,K,L*). Whole-mount immunostaining of flat-mounted retinas with Chat revealed the mosaic distribution of SACs (*Figure 7—figure supplement 2A–F*). The number of SACs in the GCL was unchanged at P15 (n = 4) and 1 month (n = 6) in *Dcc* cKO mice compared to *Dcc*<sup>fl/fl</sup> mice (*Figure 7—figure supplement 2G*; *Figure 7—source data 2*). Furthermore, no significant difference was observed in the number of SACs soma in the inner nuclear layer in *Dcc* cKO mice at P15 (n = 4) and 1 month compared to *Dcc*<sup>fl/fl</sup> controls (*Figure 7—figure supplement 2H*, *Figure 7—source data 2*; n = 6). However, sagittal cryosections of 1 month-old *Dcc* cKO retinas labeled for Chat showed that the relative thickness of the S2-S4 was significantly reduced in *Dcc* cKO mice (*Figure 7O,P,R*; n = 3) compared to *Dcc*<sup>fl/fl</sup> mice (*Figure 7I,K,L*; *Figure 7—figure supplement 2I,J*; *Figure 7—source data 2*; n = 3). There were no aberrant dendrite projections of Chat<sup>+</sup> SACs in *Dkk3:cre;Dcc*<sup>fl/fl</sup> mice compared to *Dcc*<sup>fl/fl</sup> littermates (n = 3 for each genotype).

This observation led us to question whether the overall thickness of the IPL was also affected. Horizontal cells as well as displaced amacrine cells were visualized using the calcium-binding protein Calbindin (CaBP) (*Wässle, 2004*). At 1 month, *Dcc*<sup>fl/fl</sup> retinas showed that displaced amacrine cell dendrites stratified at the border of S1-2, S2-3, and S3-4 (*Figure 7G,H,K,L*; *Wässle, 2004*). *Dkk3:*

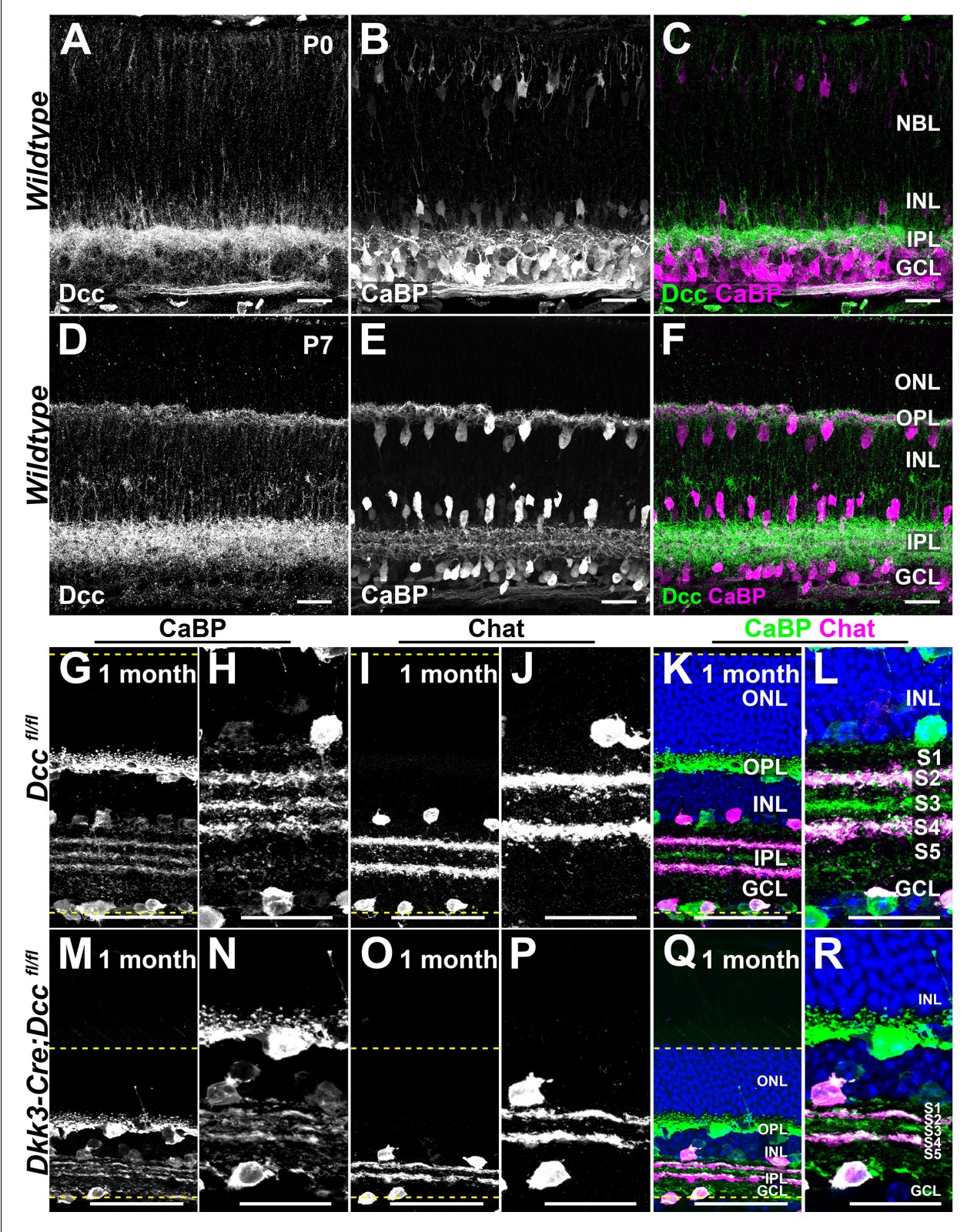

**Figure 7.** Eye-specific deletion of Dcc leads to reductions in retinal layers thickness. (**A–F**) IHC of postnatal retinas labeled for Dcc (green) and the horizontal and amacrine cell marker, Calbindin (CaBP, magenta). (**A–C**) P0 retinas. (**D–F**) P7 retinas. (**G–R**) Cryosections of 1 month-old retinas of *Dcc^{fl/fl}* and *Dkk3:cre;Dcc^{fl/fl}* mice. (**G,H,M,N**) CaBP IHC (green) that labels amacrine cells that stratify in the IPL strata S2, S3, and S4. (**I,J,O,P**) IHC for choline O-acetyltransferase (Chat) that labels starburst amacrine cells which arborize in the IPL strata S2 and S4. (**K,L,Q,R**) show merge images. Yellow dashed

*Figure 7 continued on next page*

*Figure 7 continued*

lines delineate the retinal contours. Scale bars: (**A–F,H,J,L,N, P,R**) 25 μm; (**G,I,K,M,O,Q**) 50 μm. NBL, Neuroblastic Layer; ONL, Outer Nuclear Layer; OPL, Outer Plexiform Layer; INL, Inner Nuclear Layer; IPL, Inner Plexiform Layer; GCL, Ganglion Cell Layer.

The online version of this article includes the following source data and figure supplement(s) for figure 7:

**Source data 1.** Early loss of Dcc leads to a significant and progressive degeneration of RGCs.
**Source data 2.** Eye-specific loss of Dcc does not induce a loss of SACs but impacts retinal thickness.
**Source data 3.** RGC loss is dependent on Dcc signaling.
**Figure supplement 1.** Eye-specific *Dcc* deletion leads to a dramatic reduction of RGC number.
**Figure supplement 2.** *Dcc* deletion does not affect Chat[+] cell number.
**Figure supplement 3.** Dcc signaling is critical for RGC survival.

*cre;Dcc^fl/fl* displayed a significant reduction in the IPL (n = 3) compared to *Dcc^fl/fl* mice (**Figure 7—figure supplement 2J**; **Figure 7—source data 2**; n = 3). We nonetheless found that CaBP[+] amacrine cells exhibited normal lamina-specific neurite stratification in the IPL (**Figure 7M,N,Q,R**). We analyzed the overall thickness of the retina from the outer segments of photoreceptors to the soma of RGCs. At 1 month, mutant retinas displayed a dramatic reduction in thickness compared to *Dcc^fl/fl* retinas (**Figure 7—figure supplement 2J**; **Figure 7—source data 2**; n = 3). Of note, this reduction was homogeneous across the entire retina.

To determine whether loss of RGCs was dependent on Dcc signaling, we carried out a Rbpms whole-mount immunostaining of *Dcc^Kanga* eyes (**Figure 7—figure supplement 3A,B**). At 1 month, RGC numbers were dramatically reduced by ~72% in *Dcckanga^kanga/-* mutants (n = 4) compared to the *Dcc^kanga/+* control littermates (**Figure 7—figure supplement 3C**; **Figure 7—source data 3**; n = 4). We next investigated whether the number of SACs were affected in *Dcc^kanga* mutants. In the RGC layer, SACs was not affected in *Dcckanga^kanga/-* mice (n = 4) compared to *Dcc^Kanga/+* mice (**Figure 7—figure supplement 3D,E**; n = 4). The total number of SACs in the inner nuclear layer was also unchanged in *Dcckanga^kanga/-* retinas (n = 4) compared to *Dcc^kanga/+* littermates (**Figure 7—figure supplement 3F**; **Figure 7—source data 3**; n = 4).

## Dcc deletion leads to major retinal dysplasia and visual deficits

In order to assess the overall eye phenotype in *Dkk3:cre;Dcc^fl/fl* conditional knockout mice, we adapted the EyeDISCO protocol for adult mouse eyes (see Materials and methods; **Figure 8A–B′**). This led to a modest and isotropic (rostro-caudal/medio-lateral/dorso-ventral) shrinkage of the tissue of ~11% (**Figure 8—figure supplement 1A–D**, **Figure 8—source data 2**, n = 8 eyes). By eye fundoscopy, large lesions were observed in *Dcc* cKO mutant retinas (**Figure 8C,D**). To analyze the precise localization and density of these lesions, we carried out whole-mount nuclear staining (TO-PRO-3) followed by EyeDISCO clearing and LSFM. 3D rendering showed that eye-specific *Dcc* mutants displayed abnormal conformation of photoreceptors that resembled rosette-like structures, reminiscent of photoreceptor degeneration (**Figure 8E,F**, **Figure 8—video 1**; **Chang et al., 2002**; **Flynn et al., 2014**). Rosette-like structures were specifically located in the outer nuclear layer of the retina (photoreceptor layer) and the area they covered was manually segmented (**Figure 8G,H**, **Figure 8—video 2**). This showed that unlike RGC loss, which was homogeneous in the retina, rosette structures were specifically localized as a band that spanned naso-temporally with a ventral bias (**Figure 8I–N**; n = 8). To confirm that rosettes were composed of photoreceptors, we carried out whole-mount immunolabeling for short-wavelength opsin (Opn1sw), which is known to be present in the ventral half of the retina (**Applebury et al., 2000**; **Ortín-Martínez et al., 2014**). At 1 month, Opn1sw was expressed in high-ventral and low-dorsal gradient in the *Dcc^fl/fl* eyes (**Figure 8O**; **Figure 8—video 2**; n = 6 eyes). In *Dcc* mutants, Opn1sw gradient was conserved, but rosette-like structures were observed in Opn1sw[+] photoreceptor cells (**Figure 8P**; **Figure 8—video 2**; n = 6 eyes). We next questioned whether specific types of photoreceptors would cluster in rosettes. To do so, we labeled retinas with short- and mid-wavelength opsins as well as Rod-specific opsin (Rhodopsin). All types of photoreceptors were present within rosettes (**Figure 8—figure supplement 2A–P**).

To determine whether loss of Dcc in early retinal development led to progressive photoreceptor degeneration, *Dkk3:cre;Dcc^fl/fl* mutant eyes were cleared at P15, 1 month, and 6 months (n = 5). The eyes were then manually segmented for rosette-like structures and the percentage of rosette territory covered in each retina was quantified (**Figure 8Q**). No significant differences were observed in

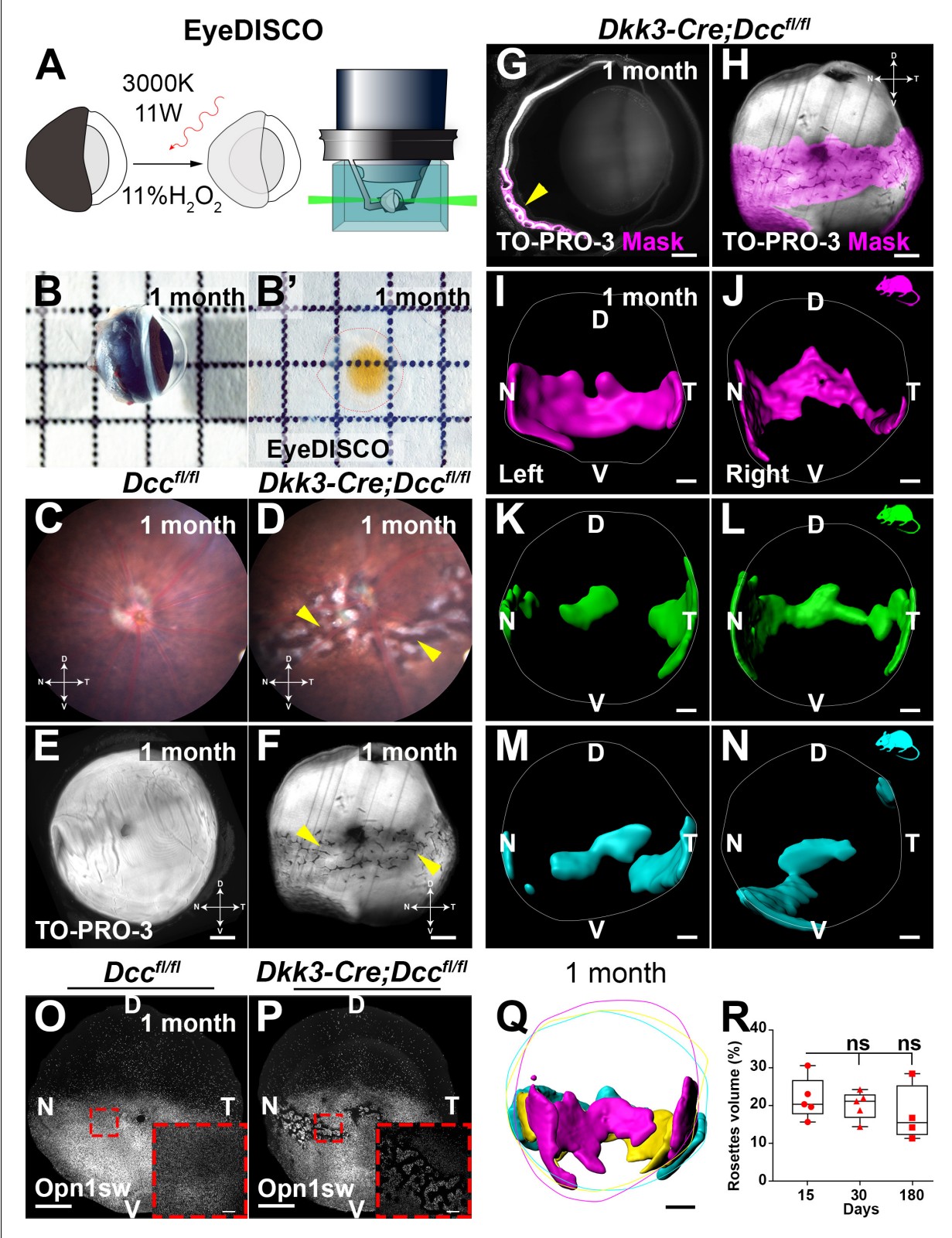

**Figure 8.** EyeDISCO, a novel tissue clearing protocol for whole eye visualization. (**A**) Is a representation of the EyeDISCO clearing pipeline. Adult eyes are dehydrated in methanol and immersed in an 11% H2O2 solution irradiated with a 3000°K warm white light (red arrow). After immuno-labeling the eyes are cleared and imaged using light sheet microscopy. (**B, B'**) Side view image of an adult eye (1 month) before and after EyeDISCO clearing. (**C,D**) Eye fundoscopy of a 1 month-old *Dcc<sup>fl/fl</sup>* and a *Dkk3:cre;Dcc<sup>fl/fl</sup>* eye. *Dkk3:cre;Dcc<sup>fl/fl</sup>* fundoscopy shows a severe dysplasia (yellow arrows). Eye

*Figure 8 continued on next page*

*Figure 8 continued*

coordinates are highlighted, D, dorsal, V, ventral,N, nasal, T, temporal. (**E,F**) Whole eye of *Dcc^fl/fl* and *Dkk3:cre;Dcc^fl/fl* mice after EyeDISCO treatment labeled with a nuclear marker (TO-PRO-3, white). Major dysplasia can still be observed in *Dkk3:cre;Dcc^fl/fl* eyes (yellow arrows). (**G**) Represents a slice of the 3D stack from a *Dkk3:cre;Dcc^fl/fl* eye. The dysplasia can be isolated by manual segmentation with Imaris (magenta, yellow arrow). (**H**) The affected area visualized in 3D (magenta) following manual segmentation. (**I–N**) 3D surfaces of rosette masks in 3 separate *Dkk3:cre;Dcc^fl/fl* mice showing both the left and right eye. (**O, P**) Whole-mount IHC of eyes of *Dcc^fl/fl* and *Dkk3:cre;Dcc^fl/fl* 1 month-old eyes labeled for short-wavelength Opsin (Opn1sw, white). (**Q**) A merge of 3 separate masks using Imaris from *Dkk3:cre;Dcc^fl/fl* mice showing rosette area coverage and retina circumference. (**R**) Quantification of rosette area coverage in retinas. No differences in rosette coverage were seen between *Dcc* cKO and control littermates at P15 (21.85 ± 2.48%; n = 5 eyes), 1 month (20.11 ± 1.67%; n = 5 eyes; p>0.999) and 6 months (17.68 ± 3.76%; n = 5 eyes; p=0.6176). A Kruskal-Wallis test was used to measure significance. Data are represented as a box plot; whiskers represent min to max values. Results were considered non-significant (ns) if p>0.05. Scale bars: (**E–H, I–N**) 300 μm, (**O, P**) 500 μm, (**O, P**) high magnification) 150 μm, (**Q**) 400 μm.

The online version of this article includes the following video, source data, and figure supplement(s) for figure 8:

**Source data 1.** Rosette volume in *Dcc* cKO mice does not progress overtime.
**Source data 2.** EyeDISCO leads to a mild and isotropic shrinkage of the adult mouse eye.
**Source data 3.** Outer nuclear layer thickness is reduced in Dcc cKo mice.
**Source data 4.** Retinal-specific deletion of Dcc leads to a significant reduction in retinal physiology.
**Figure supplement 1.** EyeDISCO leads to a mild and isotropic shrinkage of the adult eye.
**Figure supplement 2.** Photoreceptors are ubiquitously present within rosettes.
**Figure supplement 3.** Eye-specific deletion of *Dcc* leads to functional visual deficits.
**Figure supplement 4.** Dcc Kanga mutants display retinal dysplasia.
**Figure 8—video 1.** Eye-specific *Dcc* deletion leads to major retinal dysplasia.
https://elifesciences.org/articles/51275#fig8video1
**Figure 8—video 2.** Rosette formation in *Dcc* cKO mice is specific to photoreceptor cells.
https://elifesciences.org/articles/51275#fig8video2

retinas between P15, 1 month, and 6 months (*Figure 8R*; *Figure 8—source data 1*; n = 5 eyes). Thus, rosette structures in mutant mice do not progress over time. We then investigated whether photoreceptors outside of rosettes were also affected. To address this, we carried out an immuno-labeling on 1-month-old retinas with the pan-photoreceptor marker, Recoverin. We observed that in *Dcc* cKO retinas (n = 3) photoreceptors were dramatically reduced compared to *Dcc^fl/fl* controls (n = 3) (*Figure 8—figure supplement 2Q–U*; *Figure 8—source data 3*).

Finally, we also observed that rosettes induce a significant loss in physiological visual response. Electroretinogram (ERG) at 1 month showed that the scotopic response of *Dcc* cKO mice was significantly reduced in a-wave but not b-wave complexes (*Figure 8—figure supplement 3A*; *Figure 8—source data 4*). This reduction persisted at 6 months in a-wave but not b-wave complexes (*Figure 8—figure supplement 3A*). To assess cone function, we also analyzed the photopic response in *Dcc* cKO mice and found a dramatic loss of photopic response at 1 and 6 months (*Figure 8—figure supplement 3B*; *Figure 8—source data 4*). Altogether, eye-specific deletion of *Dcc* leads to rosette formation in the photoreceptor layer of the retina. These rosettes are localized to the naso-temporal retina and do not progress overtime.

Strikingly, as was observed in our *Dcc* cKO mice, rosettes were present in *Dcc Kanga* mutants as early as P15 suggesting a developmental defect (*Figure 8—figure supplement 4E–F*; n = 4 eyes). These rosettes were still present at 1 month (*Figure 8—figure supplement 4F–G*; n = 9 *Dcc^kanga/-* and n = 10 *Dcc^kanga/+*). Both short- and mid-wavelength cones were present within the rosettes (*Figure 8—figure supplement 4E–G*). Therefore, altered Dcc signaling leads to the formation of rosettes in the outer nuclear layer of the retina.

## Discussion

### Dcc is essential for RGC intraretinal axon guidance

Early eye development begins from the invagination of the optic vesicle to form the optic cup (*Bernstein et al., 2018*). This morphogenetic event gives rise to the neural retina and forms an opening along the ventral midline that will become the future optic disc, also known as the choroid fissure (*Bernstein et al., 2018*). Early RGCs are polarized and extend within the basal retina, constituting the optic fiber layer (OFL). RGC axons further extend circumferentially towards the presumptive

optic disc where they exit the retina (*Bao, 2008*). Therefore, two mechanisms are at play during the intraretinal navigation of RGC axons.

The first process involves the basal lamina at the vitreal side which provides a growth-promoting substrate for RGC axons. Several guidance cues such as Slits and Semaphorin3E restrict RGC axons to the OFL (*Jin et al., 2003*; *Steinbach et al., 2002*; *Thompson et al., 2006*). However, the basal lamina does not provide directionality (*Halfter et al., 1987*; *Halfter and Fua, 1987*) and other guidance cues such Sfrp1, Sfrp2, EphB2, EphB3, Netrin-1 present in the retina (*Birgbauer et al., 2000*; *Deiner et al., 1997*; *Marcos et al., 2015*), and Slit2 present in the lens (*Thompson et al., 2006*) orient RGC axons towards the optic disc.

Interestingly, pioneer RGC axons extend into the optic vesicle during choroid fissure closure (*Deiner et al., 1997*; *Kuwabara and Weidman, 1974*). Perturbations of this event have been associated with optic nerve defects (*Cai et al., 2013*; *Dakubo et al., 2003*; *Morcillo et al., 2006*; *Otteson et al., 1998*; *Silver and Robb, 1979*). It was recently shown that Netrin-1 is required for choroid fissure closure, and that *Ntn1*$^{-/-}$ mice display highly penetrant colobomas (*Hardy et al., 2019*). We observe that our Ntn1 cKO also exhibit colobomas (80% penetrance, n = 5), suggesting that Netrin-1 at the optic disc is required for proper choroid fissure closure. We could not detect any choroid fissure defects in either *Dcc* cKO or *Dcc* null mutant mice suggesting that Netrin-1 may mediate choroid fissure fusion in a Dcc-independent manner.

The first born RGCs arise in the dorso-central retina in close proximity to the pax2-positive optic disc cells (*Drager, 1985*) expressing Netrin-1 (*Deiner et al., 1997*). Interestingly, we observe no significant differences between the optic nerve hypoplasia displayed in Netrin-1 null mice and Ntn1 cKO mice. Thus, as it was described in the hindbrain and the spinal cord (*Dominici et al., 2017*; *Moreno-Bravo et al., 2019*; *Varadarajan et al., 2017*; *Wu et al., 2019*), it is likely that Netrin-1 acts intraretinally in a short-range manner.

## Dcc is necessary for retinal ganglion cell projections in the primary and accessory optic systems

RGC projections initially reach the thalamus and the superior colliculus during embryonic development (*Godement et al., 1984*). Different axon guidance molecules are critical for the proper targeting of RGC projections to the thalamus and to the superior colliculus. In the absence of reelin, RGC axons fail to target the vLGN and the IGL correctly (*Su et al., 2011*). Once RGC axons have reached their target they then refine between ipsi- and contra-lateral territories. During this process, EphB or ephrinBs are required for RGCs to dictate their final position in the superior colliculus (*Hindges et al., 2002*; *Thakar et al., 2011*). In the *Dcc* cKO, RGCs target the LGN and superior colliculus normally but we observe an expansion of the ipsilateral territory, thus hinting at a possible refinement defect. Netrin-1 and Dcc have been shown to play a role in RGC axon arborization and synapse formation within the optic tectum of *Xenopus* tadpoles (*Manitt et al., 2009*). Here, we show for the first time that intra-retinal loss of Dcc leads to an expansion in ipsilateral territory in the murine SC. Nevertheless, we cannot rule out that the expansion in ipsilateral projections within the SC may be a result of the significant death of RGCs observed in *Dcc* cKO retinas.

The organization of the accessory optic system (AOS) projections to the MTN, the NOT, and the OPT is altered in *Dcc* mutants. These defects were constant and reproducible. Moreover, Alexa-Fluor-CTB is fluorescent enough to label isolated axons (*Li et al., 2015*) and therefore it is unlikely that a significant fraction of visual axons could not be traced. Although the major input to the AOS is constituted of ON Direction Selective Ganglion Cells (DSGC) (*Dhande et al., 2013*), ON-OFF DSGCs also project to AOS nuclei (*Kay et al., 2011*). The MTN is innervated by ON-OFF DSGCs which control eye movement and give information for upward motion (dMTN) or downward motion (vMTN) (*Borst and Euler, 2011*; *Yonehara et al., 2009*). However, RGCs projecting to the MTN are homogeneously spread across the retina (*Sun et al., 2015*). Little is known about the cues guiding DSGC axons to the MTN besides the transmembrane semaphorin Sema6A and its receptors PlexinA2/A4 (*Sun et al., 2015*). Contactin-4 and the amyloid precursor protein are necessary for DSGC's axons to project in the NOT, but not the MTN (*Osterhout et al., 2015*). As Dcc protein is homogeneously expressed in embryonic RGCs, one could wonder why we observe a specific guidance defect in the MTN. One hypothesis could be that co-receptors of Dcc, such as Unc5d could potentiate the MTN defect observed in our *Dcc* mutants (*Murcia-Belmonte et al., 2019*). Furthermore, the differential distribution of Netrin-1 protein within the visual nuclei could be another area of study to

explain the heterogeneous defects observed in our Dcc mutants. Still, we cannot rule out that the loss of RGCs in our Dcc mutants could impact axonal arborization due to a reduced competition between neighboring RGC axons. Further investigations should be performed to test whether DSGCs may be specifically impacted by Dcc perturbation using MTN-specific mouse lines such as the SPIG1:GFP (*Yonehara et al., 2008*) and Hoxd10:GFP (*Dhande et al., 2013*) knock-in mouse lines. Overall, we uncover a new role for Dcc in the innervation of the MTN, the NOT and the OPT.

## Why do RGCs die in eye-specific *Dcc* mutants?

Although the overall death of RGCs is homogeneous, only a portion aberrantly invade the subretinal space. This suggests that the death of RGCs is not merely correlated to the misprojection phenotype. To survive, RGCs require trophic factors such as brain-derived neurotrophic factor (BDNF), ciliary neurotrophic factor (CNTF), neurotrophin 4 (NT4), and fibroblast growth factor (FGF) secreted by the superior colliculus (*Meyer-Franke et al., 1995*; *Raff et al., 1993*). A possibility is that too few RGC axons make it to the superior colliculus to provide sufficient trophic support. This massive RGC death could also result from the elimination of the misrouted RGCs axons as previously demonstrated in the zebrafish (*Poulain and Chien, 2013*). Netrin-1 through Dcc has been shown to promote synaptogenesis of cortical neurons (*Goldman et al., 2013*). Therefore, an abnormal synaptogenesis of *Dcc*-deficient RGC axons with their target neurons could also induce their degeneration.

## Photoreceptor cell degeneration

In addition to RGCs death *Dcc* cKO mutants display major retinal dysplasia, limited to the retinal outer nuclear layer. These undulations in the ONL often referred as rosettes in the literature (*Mears et al., 2001*), are among the most prominent features of retinal degeneration along with photoreceptor cell death and underdeveloped outer segments (*Genové et al., 2014*). In *Dkk3:cre; Dcc^lox/lox* mice, the rosettes are only present in the inferior half of the retina but span from the nasal to the temporal sides. To our knowledge, localized rosettes have only been described in *Crb1* mutant mice. *Crb1^rd8/Crb1^rd8* mice display rosettes in the inferior nasal quadrant of the retina (*Mehalow et al., 2003*) whereas in *Crb1^−/−* mice retinal degeneration occurs in the inferior temporal quadrant (*van de Pavert et al., 2004*). Importantly, *Dcc* cKO mice have been backcrossed into a C57BL/6J background and are negative for the *rd10* mutations (*Supplementary file 3*). What could explain the degeneration of a subset of photoreceptors? Based on our results, it is more likely that this phenotype is related to the intra-retinal pathfinding defects. First, rosettes were observed as early as E15, concomitantly with the first misrouted axons in the retina. Therefore, misguided RGC axons invading the subretinal space could separate photoreceptors from RPE. Indeed, interactions between the RPE and presumptive photoreceptor outer segments are crucial for their proper development (*Marmorstein, 2001*). Early ablation (E10–11) of RPE in mouse retina also results in a disorganization of retinal layers (*Raymond and Jackson, 1995*). Moreover, rats with dysfunctioning RPE cells show photoreceptor death (*D'Cruz et al., 2000*). Second, several studies have linked rosette formation with outer limiting membrane defects (*Mehalow et al., 2003*; *Rich et al., 1995*; *Stuck et al., 2012*). It is thus likely that misguided RGC axons in our *Dcc* cKO mice alter the outer limiting membrane leading to photoreceptor degeneration. Third, we show that Dcc is expressed homogeneously in Crx[+] cells at E15. Using the *Dkk3* promoter leads to a complete removal of *Dcc* in all retinal progenitors. If Dcc played a role in proper photoreceptor development we would expect that all photoreceptor cells would degenerate. Finally, we observe that M- and S-cones as well as rods are all present within rosettes. These results support the idea that in *Dcc* cKO mice, photoreceptor cell death is not limited to a specific cell type but rather triggered by RGC axons. Nonetheless, retinal physiology in our Dcc mutants is globally perturbed, indicating either that rosettes (which account for ~20% of the retina) are sufficient to impact the ERG or that photoreceptor defects may not only be restricted to rosette territories. Quantification of the ONL outside of rosettes in Dcc cKO mice shows a dramatic thinning compared to control littermates. It would be of interest to investigate whether Dcc plays a cell autonomous role in photoreceptors.

## Dcc's mechanism of action

Here, we show that Dcc acts cell autonomously in RGCs to confine their axons to the retina but that the phenotype is heterogeneous. In *Dkk3:cre;Dcc^fl/fl* mice, some RGC axons exit the retina and even project to the CNS, suggesting that they are still properly guided in the absence of Dcc. On the other hand, other RGC axons aberrantly project apically into the subretinal space. These results suggest that the long-range guidance of RGC axons to the optic disc is not mediated by Dcc. It rather acts as a short range guidance receptor preventing RGC axons from escaping the neural retina and staying in the OFL, as shown for hindbrain commissural axons (*Moreno-Bravo et al., 2018*; *Yung et al., 2018*). Another in vivo example of the short-range guidance role of Dcc/Netrin-1 signaling was shown in R8 photoreceptor growth cones in *Drosophila melanogaster*, which could successfully project but not attach properly in their terminal zone (*Akin and Zipursky, 2016*). Why the phenotype is heterogeneous is unclear as the expression levels of Dcc (based on immunocytochemistry) in embryonic RGCs appear similar across the retina. This could be explained by a differential expression of co-receptors or downstream partners in different subclasses of RGCs. Evidence for this phenotypic heterogeneity is also found at a later stage and further suggests that specific populations of RGCs are differentially affected by the loss of Dcc.

Most of Dcc's known functions such as growth cone attraction are mediated by its P3 intracellular domain (*Finger et al., 2002*). The *Dcc^Kanga* mice lacking the exon encoding the P3 intracellular domain present the same phenotype as the *Dcc* KO mice, with a missing corpus callosum and an aberrantly projecting corticospinal tract (*Finger et al., 2002*). The observation that *Dcc^Kanga* mice share many of the phenotypes observed in *Dcc* cKO mice suggests that they are indeed directly dependent on Dcc signaling and not the signaling of a Dcc co-receptor. Moreover, we show that deletion of Netrin-1 phenocopies the defects observed in the embryonic retina. Thus, we demonstrate that Dcc/Netrin-1 signaling is critical for the proper targeting of RGCs.

## EyeDISCO is a powerful technique to study the eye

In the past decade, there have been significant developments in tissue clearing protocols. However, none have addressed the challenge of clearing densely pigmented tissues (*Susaki and Ueda, 2016*). Recent protocols allowing for whole-mouse visualization are still hindered by densely pigmented tissues such as the eyes or the skin (*Cai et al., 2019*; *Pan et al., 2016*; *Tainaka et al., 2018*). Efforts to remove densely pigmented tissues such as the melanophores of the RPE of the eye have almost entirely been carried out on tissue sections (*Alexander et al., 1996*; *Orchard, 2007*). For instance, using potassium permanganate and oxalic acid in aqueous conditions Iwai-Takekoshi and colleagues were able to remove RPE pigments in embryonic sections (*Iwai-Takekoshi et al., 2016*). This depigmentation protocol has however been shown to reduce the compatibility/efficacy of immunolabelling (*Alexander et al., 1996*). Recently, a de-pigmentation protocol of postnatal eyes amenable to tissue clearing has been published (*Henning et al., 2019*). This protocol carries out an $H_2O_2$ treatment in aqueous conditions which favors the generation of microbubbles that damage the retina and result in retinal detachment (*Henning et al., 2019*). In addition, the $H_2O_2$ treatment is done at 55°C, further deteriorating as well as reducing antigenicity of the tissue. Only 2 antibodies were validated using this protocol. Here, we report an amenable tissue clearing protocol for whole-embryo and whole adult eye clearing compatible with immunolabeling. The immunolabeling protocol is carried out at RT and requires passive diffusion of low titer antibodies. Thus far, 17 antibodies have been validated, encompassing transcription factors, transmembrane and cytoplasmic proteins (*Supplementary file 2*).

A current limitation of our method is the cellular resolution at which we are able to carry out whole-eye imaging using LSFM, as our setup is composed of a macroscope with a 2X objective coupled to a 6.3X numerical zoom (12.6X) giving a maximum x/y resolution of 5.16 μm/pixel. However, the recent development of higher magnification objectives compatible with DBE will allow for more resolved images, equivalent to confocal microscopes, that enable whole-eye cell-counting approaches (*Pan et al., 2019*).

Altogether, EyeDISCO provides a novel opportunity to image the visual system like never achieved before. This protocol allows the study of visual projections from the eye to the brain, thereby constituting a powerful tool for the screening of mutants. This protocol could also prove extremely useful to study the eye as a whole for better understanding eye disorders.

## Materials and methods

### Ethics statement

All experiments were designed using the 3R rule: to reduce, refine, and replace the use of animals. All animal procedures were carried out according to approved institutional guidelines (#B-75-12-02) of the Institut de la Vision. The protocol was approved by the Sorbonne Université ethic committee (Charles Darwin) (Permit Number: 9571). In cases of animal handling, experiments were performed to minimize animal stress and suffering.

### Animals

*Dcc* knockout mice (*Fazeli et al., 1997*), *Dcc*$^{kanga}$ mice (*Dcc*$^{kanga}$; Jackson Laboratories) (*Finger et al., 2002*), *Dcc* floxed mice (*Krimpenfort et al., 2012*), *Ntn1* floxed mice (*Moreno-Bravo et al., 2018*), and *Dickkopf3:cre* (*Dkk3:cre*) mice (*Sato et al., 2007*) were previously described. All mice are kept in C57BL/6J background. Cre expression was monitored by crossing *Dkk3:cre* mice with the Ai9 Rosa$^{tdTomato}$ reporter line (Rosa$^{Tom}$; Jackson Laboratories). The day of the vaginal plug was counted as embryonic day 0.5 (E0.5). Mice were anesthetized with ketamine (80 mg/kg) (Axience) and xylazine (8 mg/kg) (Axience). Embryos and adult mice of either sex were used.

### RD gene sequencing

PDE6b$^{rd1}$ mutations were investigated by direct Sanger sequencing. Genomic DNA was extracted using 50 mM NaOH for 30 min at 95°C. The following primers were used Forward: 5'ctgcacacaga-catccagtc3' Reverse: 5'ccatgcctggctgaagttgt3'. PCR was done using a Gotaq DNA Polymerase (Promega). Next, PRC products were sequenced using Sanger sequencing (BigDyeTermv1.1 Cycle-Seq kit, Applied Biosystems) and analyzed on an automated 48-capillary sequencer (ABI3730 Genetic analyzer, Applied Biosystems), and the results interpreted by applying a software (Seqscape, Applied Biosystems).

### Immunohistochemistry

Embryos were fixed by immersion in 4% paraformaldehyde in 0.12 M phosphate buffer (VWR, 28028.298 and 28015.294), pH 7.4 (PFA) o/n at 4°C. Eyes were harvested and using a 30$^{1/2}$G needle, a small hole was made in the cornea. The eyes were then fixed in 4%PFA for 1 hr at RT. Following three washes in 1XPBS, the samples were incubated in 10% sucrose (VWR, 27478.296) in 0.12 M phosphate buffer o/n at 4°C. The next day, samples were transferred to a 30% sucrose solution in 0.12 M phosphate buffer o/n at 4°C. Samples were then embedded in 0.12 M phosphate containing 7.5% gelatin (Sigma, 62500) and 10% sucrose, frozen in isopentane at −50°C and then cut at 20 μm with a cryostat (Leica, CM3050S). Sections were blocked in PBS containing 0.2% gelatin (VWR) and 0.25% Triton-X100 (PBS-GT) for 1 hr at RT. Sections were then incubated with primary antibodies (see *Supplementary file 1*) diluted in a PBS-GT solution o/n at RT. Following three washes in PBST (0.05% Triton-X100) secondary antibodies coupled to the appropriate fluorophore (see *Supplementary file 1*) were diluted in PBS-GT and incubated for 2 hr at RT. Sections were counter-stained with Hoechst (Sigma, B2883, 1:1000). Slides were scanned with either a Nanozoomer (Hamamatsu) or laser scanning confocal microscope (Olympus, FV1000). Brightness and contrast were adjusted using ImageJ.

### Retinal thickness

To measure the thickness of retinal layers, 1 month-old control and *Dkk3:cre;Dcc*$^{fl/fl}$ retinas were harvested and fixed in 4% PFA o/n at 4°C. Retinas were processed as previously described. Calbindin (Swant, see *Supplementary file 1*) and Chat (Millipore, see *Supplementary file 1*) immunostaining were used to visualize the IPL layers and Hoechst (Sigma) to visualize all retinal layers. Using NDP viewer (version 2.2.6), 5 retinal sections from the optic nerve were analyzed (n = 3 mice per group). Using the measure tool, the thickness (μm) of each layer was measured.

## EdU proliferation

Pregnant females were injected intraperitoneally with 5-ethynyl-2′-deoxyuridine (EdU;1 mg/10 g) and sacrificed 3 hr following injection. Proliferating cells were visualized using the Click-iT EdU Imaging kit (Invitrogen) and were co-labeled with antibodies against Chx10 (Exalpha, see *Supplementary file 1*), a retinal progenitor cell marker. For co-localization of EdU/Chx10, the Imaris X 64 software (Bitplane, version 9.1.2) co-localization tool was used and the percentage of ROI co-localization was measured (threshold Chx10 = 320, threshold EdU = 147).

## Tracing of visual projections

Mice were anesthetized with an intra-peritoneal injection of ketamine (100 mg/kg) and xylazine (10 mg/kg) and kept warm with a thermostatically controlled platform at 37°C. Corneal analgesia was done by applying chlorhydrate oxybuprocaine (1.6 mg/0.4 ml). The eye was proposed using bulldog forceps (FST, 18039–45). A $30^{1/2}$-gauge needle was used to make a pre-hole at the dorsal side of the eye. Using a Nanofil syringe (World Precision Instruments, Nanofil) with a $33^{1/2}$–gauge beveled needle (World Precision Instruments, NF33-BV2), 1.2 µl 2 µg/µl of AlexaFluor-conjugated cholera toxin β subunit (Thermo Fischer, AlexaFluor555-CTB C22843 and AlexaFluor647-CTB C34778) was injected intravitreally. To avoid leakage, the needle was slowly withdrawn over the span of 3 s. 72 hr following CTB injection, mice were transcardially perfused with 4%PFA and the brains were dissected for tissue clearing.

## Flat mount

For retinal flat mounts, eyes were harvested and the retina were dissected and fixed by gentle shaking at 50 rpm in 4%PFA for 45 min at RT. The retinas were then washed three times in 1XPBS. For immunohistochemistry, retinas were permeabilized and blocked in a solution containing 0.5% Triton-X100, 5% donkey normal serum, 1XPBS, 0.1 g/L thimerosal for 1 day at RT under agitation. Primary antibodies (see *Supplementary file 1*) were diluted in a solution containing 0.5% Triton-X100, 5% donkey normal serum, 10% Dimethyl Sulfoxide, 1XPBS, 0.1 g/L thimerosal for 3 days at RT under agitation. The retinas were then washed for 1 day in PBST (1XPBS, 0.5% Triton-X100). The secondary antibodies (see *Supplementary file 1*) were diluted in the same solution as primary antibodies and left for 2 days. After washing retinas for 1 day, they were mounted on slides and imaged using a scanning confocal microscope (Olympus, FV1000).

For quantifications of Rbpms$^+$ (Phosphosolutions, see *Supplementary file 1*) and Chat$^+$ (Millipore, see *Supplementary file 1*) cells, the Imaris X 64 'Spots' tool was used (Bitplane, version 9.1.2). For Rbpms$^+$ staining, automatic segmentation was done. For Chat staining, a manual segmentation was carried out to separate the inner nuclear layer from the retinal ganglion cell layer. Automatic cell counting was then carried out on each segmentation.

## Electroretinogram

All experiments were carried out in double-blind. Following o/n adaptation, animals were prepared under red light and were anesthetized with an intraperitoneal injection of ketamine (80 mg/kg) (Axience) and xylazine (8 mg/kg) (Axience). Pupils were dilated with 0.5% tropicamide (CSP) and 5% neosynephrine (CSP, France). Corneal analgesia was performed by applying chlorhydrate oxybuprocaine (CSP). Eyes were proposed using bulldog forceps (FST, 18039–45). Recording small gold loop electrode contacting the cornea through a layer of Lubrithal (Centravet) was used to record the retinal response, with needle electrode placed in the head and back used as the reference and ground electrodes, respectively. Body temperature was maintained at 37°C with a heating pad. Electroretinograms (ERGs) were obtained simultaneously from both eyes, the light stimulus was provided by Led in a Ganzfeld stimulator (Espion, Diagnosys LLC). Scotopic responses were measured in darkness, during flash stimulation (0.003 to 10 cd.s/m$^2$), a flash duration of 4 ms. Scotopic ERG response is the mean of five responses. Photopic cone ERGs were performed on a rod-suppressing background after 5 min of light adaptation (2 cd.s/m$^2$); recordings were obtained at light intensities of 10 cd.s/m$^2$. Photopic ERG response is the mean of ten responses. Responses were amplified and filtered (1 Hz-low and 300 Hz-high cutoff filters) with a one-channel DC-/AC- amplifier.

## Eye fundoscopy

Eye fundoscopy was carried out using (MicronIV, Phoenix Research Labs, USA). Pupils were dilated with 0.5% tropicamide (CSP) and 5% neosynephrine (CSP). Next, mice were anesthetized with 5% isoflurane inhalation (Axience) and maintained at 2% Lubrithal (Centravet) was used to protect the cornea during acquisition. To visualize blood vessels in vivo, an 0.1% sodium fluorescein tracer (Serb) was injected intraperitoneal and eye fundoscopy was carried out.

## Whole-mount labeling and tissue clearing

### EyeDISCO

### De-pigmentation

For samples E12-E16, samples were fixed o/n at 4°C in 4% PFA. The embryos were then de-hydrated in succeeding baths of methanol for 2 hr each at RT (40% 1XPBS, 80% distilled water (dH$_2$O), 100% methanol). Samples were then placed o/n in a de-pigmentation solution of methanol containing 11% H$_2$O$_2$ (VWR, 216763) at 70 rpm exposed to an 11W warm white Light-Emitting Diode (LED) (3000° Kelvin).

For samples E16-P7, samples were processed as described above. However, the de-pigmentation solution was refreshed twice per day to ensure full activity. For complete de-pigmentation approximately 2 days were required.

For samples P7-Adult, samples were processed as described above. For complete de-pigmentation approximately 5 days were required.

Once completely de-pigmented, samples were gently re-hydrated in 2 hr baths at RT (100% methanol, 80% dH$_2$O, 40% 1XPBS, 1XPBS). Samples were kept at 4°C for further processing.

### Whole-mount immunostaining

For eyes after P15, an incision was done in the cornea (1/4 of the perimeter). Samples were then permeabilized in the blocking solution (0.5% Triton-X100, 5% donkey normal serum, 1XPBS, 0.1 g/L thimerosal) for 1 day at RT on agitation. For immunostaining, samples were incubated with the primary antibodies (see *Supplementary file 1*) in a solution containing: 0.5% Triton-X100, 5% donkey normal serum, 20% Dimethyl Sulfoxide, 1XPBS, 0.1 g/L thimerosal. The primary antibody solution (see *Supplementary file 1*) was incubated for 7 days at RT. The samples were then washed for 1 day (6 changes) in PBST (0.5% Triton-X-100, 1XPBS, 0.1 g/L thimerosal). The secondary antibody (see *Supplementary file 1*) was diluted in the same solution as for the primary and passed through a 0.22 μm filter and incubated for 2 days in solution at RT under agitation. In some cases, samples were counterstained with the nuclear marker TO-PRO-3 (Life Technologies, T3605, 1:300). The samples were then washed for 6 times during 1 day in PBST, and 2 washes of 1XPBS prior to storing the samples in the dark at 4°C until clearing.

### Agarose embedding

Embryos (E12) and eyes were embedded in 1.5% agarose (Roth) in 1X TAE (Life Technologies) prior to tissue clearing.

### Tissue clearing

The iDISCO+ protocol was adapted. All steps were carried out in the dark in a fume hood by agitation at 10 rpm (SB3 tube rotator, Stuart) at RT using 15 ml centrifuge tubes (TPP). Following embedding, eyes were placed in 20% methanol diluted in 1XPBS o/n. The next day, eyes were de-hydrated in succeeding baths of methanol for 2 hr (40% 1XPBS, 60% 1XPBS, 80% dH$_2$O, 100% methanol). The eyes were then placed in a solution containing 2/3 Dichloromethane (DCM, Sigma) 1/3 methanol o/n. The next day, eyes were placed in DCM for 30 min prior to being immersed in the imaging medium, Di-benzyl Ether (DBE, Sigma). The next day, samples were stored in individual light-absorbing glass vials (Roth) at RT.

## 3D imaging

### Light sheet microscopy

All imaging was carried out as previously described *Belle et al. (2017)*; *Belle et al. (2014)*. Acquisitions were performed by using an ultramicroscope I (LaVision BioTec, Miltenyi Biotec) with the

ImspectorPro software (LaVision BioTec, Miltenyi Biotec, 5.1.328 version). The light sheet was generated by a laser (wavelength 488, 561, 64, or 780 nm, Coherent Sapphire Laser, LaVision BioTec, Miltenyi Biotec) and a cylindrical lens for large working distance. A binocular stereomicroscope (Olympus, MXV10) with a 2x objective (Olympus, MVPLAPO) was used at different magnifications (0.63x, 1x, 1.25x, 1.6x, 2x, 2.5x, 3.2x, 4x, and 5x). Samples were placed in an imaging reservoir made of 100% quartz (LaVision BioTec, Miltenyi Biotec) filled with DBE and illuminated from the side by the laser light. A Zyla SCMOS camera (Andor, Oxford Instrument, 2,048 × 2048 pixels size) was used to acquire images. The step size between each image was fixed at 1 or 2 µm (NA = 0.5, 150 ms time exposure). All tiff images are generated in 16-bit.

### Confocal microscopy

For 3D imaging using the confocal microscope (Olympus, FV1000), homemade chambers were created. Two ¼" stainless steel washers were stacked and glued (Best Klebstoffe) on a SuperFrost slide (ThermoScientific) and left to dry o/n at RT. The next day, the washers were sealed using dental cement (Dentalon Plus, R010024) and were left to dry for 2 hr at RT. The samples were then placed in the chambers and covered with DBE, a coverslip was then placed to secure the samples and was sealed using dental cement and left to dry for 2 hr at RT prior to imaging. For imaging, a scanning upright confocal microscope (Olympus, FV1000) was used with a 25X objective (Olympus, XLPLN25XSVMP2, NA = 1.0, WD = 4 mm).

## Image processing

3D rendering of light sheet and confocal stacks were converted to an Imaris file (.ims) using ImarisFileConverter (Bitplane, 9.1.2 version or 9.2.1 version) and then visualized using the Imaris x64 software (Bitplane, 9.1.2 version or 9.2.1 version).

To isolate the visual projections of E16 embryos labeled for Tag1 (R and D systems, see *Supplementary file 1*), a manual segmentation of the retina, optic nerve, optic chiasm, and optic tracts was carried out. Optic nerve volumes were calculated by creating an automatic segmentation (x = 545, y = 300, z = 200, surface detail = 3.02 µm, automatic threshold) and volumes were extracted from the surface. For analyzing aberrant RGC projections, a manual segmentation was carried out to isolate RGC projections. This mask was then extracted by adjusting the outside pixels to 0 (black). To further isolate aberrant RGC projections, normal projections (inside the retina) were manually segmented and outside pixels were set to 0. Aberrant RGC projections were isolated by excluding normal RGC projections and a mask was generated and pseudo-colored.

For P15 and 1 month CTB-traced mouse brains, visual nuclei were segmented as follows. For optic nerves the surface was extracted by creating an automatic surface in a region of x = 545, y = 300, z = 200, surface detail = 8.13 µm and automatic threshold. For optic tracts, x = 526, y = 437, z = 797, surface detail = 8.13 µm and automatic threshold. For contra-lateral superior colliculi, x = 640, y = 640, z = 640, surface detail = 8.13 µm and automatic threshold. For segmentation of the optic nerve, optic tracts and contra-lateral superior colliculi, the semi-automatic surface was used with a surface details fixed at 8.13 µm and an automatic thresholding. For the contralateral and ipsilateral lateral geniculate nuclei, an automatic segmentation was applied using a set voxel box (x = 337, y = 265, z = 761), surface detail = 2.00 µm. The medial terminal nuclei were segmented using automatic segmentation using a set voxel box (x = 262, y = 232, z = 540) with a surface detail = 2.00 µm. Ipsilateral superior colliculi were manually segmented using the 'isoline' tool, with a reduced density at 10%. Once the structure was segmented, a surface was generated and the volume was extracted for further analysis.

For adult retinas, rosette-like structures were manually segmented and surfaces were generated. Surfaces were then pseudo-colored and aligned to each to the appropriate dorsal/ventral/nasal/temporal coordinates. To analyze rosette evolution, rosettes and retinas were separately segmented at P15, 1 month and 6 months. A surface was generated and the volume extracted. A ratio of rosette per retinal volume was then calculated.

Movies were generated using the animation tool on Imaris x64 software (Bitplane, version 9.1.2) and movie reconstruction with. tiff series were done using ImageJ (1.50e, Java 1.8.0_60, 64-bit). All movie editing (text and transitions) was done using iMovie (Apple Inc, version 10.1.1).

## Statistical analyses

An observer blinded to the experimental conditions performed all the quantifications. All data are represented as mean values ± SEM. Statistical significance was estimated using two-tailed unpaired tests for non-parametric tendencies (Kruskall-Wallis or Mann-Whitney), two-way ANOVA and Bonferroni's multiple comparison test. *=$p < 0.05$; **=$p < 0.01$; ***=$p < 0.001$, ****=$p < 0.0001$. All statistical measurements were carried out using GraphPad Prism 7.

## Acknowledgements

We thank Dr Alexandra Rebsam for helpful discussions. We thank the phenotyping facility, in particular Julie Dégardin and Manuel Simonutti for assistance with the eye fundoscopy and electroretinograms. We are also thankful to all the dedicated staff of the Vision Institute animal house facility and to Alexis Guerin in particular. We thank Dr Stephane Fouquet (Vision Institute Imaging Facility) for technical assistance. We thank Dr Anton Berns for providing the *Dcc* conditional knockout line. The *Dkk3:cre* line was provided by the RIKEN BRC through the National Bio-Resource Project of the MEXT, Japan. This work was supported by a grant from the LABEX LIFESENSES (reference ANR-10-LABX-65) supported by French state funds managed by the Agence Nationale de la Recherche (ANR) within the Investissements d'Avenir programme under reference ANR-11-IDEX-0004–02 (AC). The funders had no role in study design, data collection and analysis, decision to publish, or preparation of the manuscript.

## Additional information

### Funding

| Funder | Grant reference number | Author |
| --- | --- | --- |
| Fondation ARC pour la Recherche sur le Cancer | DOC20190508735 | Robin J Vigouroux |
| Agence Nationale de la Recherche | ANR-11-IDEX-0004-02 | Alain Chédotal |
| Agence Nationale de la Recherche | ANR-10-LABX-65 | Alain Chédotal |

The funders had no role in study design, data collection and interpretation, or the decision to submit the work for publication.

### Author contributions

Robin J Vigouroux, Data curation, Formal analysis, Investigation, Visualization, Methodology; Quénol Cesar, Data curation; Alain Chédotal, Conceptualization, Formal analysis, Supervision, Funding acquisition, Validation, Investigation, Methodology, Project administration; Kim Tuyen Nguyen-Ba-Charvet, Conceptualization, Data curation, Formal analysis, Supervision, Validation, Investigation, Methodology, Project administration

### Author ORCIDs

Robin J Vigouroux (iD) https://orcid.org/0000-0002-3217-895X
Alain Chédotal (iD) https://orcid.org/0000-0001-7577-3794
Kim Tuyen Nguyen-Ba-Charvet (iD) https://orcid.org/0000-0001-5398-0872

### Ethics

Animal experimentation: All experiments were designed using the 3R rule: to reduce, refine, and replace the use of animals. All animal procedures were carried out according to approved institutional guidelines (#B-75-12-02) of the Institut de la Vision. The protocol was approved by the Sorbonne University ethic committee (Charles Darwin)(Permit Number: 9571). In cases of animal handling, experiments were performed to minimize animal stress and suffering.

Decision letter and Author response
Decision letter https://doi.org/10.7554/eLife.51275.sa1
Author response https://doi.org/10.7554/eLife.51275.sa2

## Additional files

### Supplementary files

- Supplementary file 1. List of primary and secondary antibodies used for the study.
- Supplementary file 2. List of primary antibodies validated with the EyeDISCO clearing protocol.
- Supplementary file 3. Sequencing of Phosphodiesterase 6b, cGMP, rod receptor, beta polypeptide (Pde6b) mutations in the *Dcc* cKO and *Dcc kanga* mice.
- Supplementary file 4. Key resources table.
- Transparent reporting form

### Data availability

All data generated or analysed during this study are included in the manuscript and supporting files. All source files are provided.

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
