## [Decision Letter]

**Acceptance summary:**

This manuscript re-assessing the function of DCC during retinal development using a new eye-specific Dcc mutant mouse (Dkk-Cre) that removes Dcc specifically in retinal progenitors, thereby bypassing lethality at birth. The authors confirm previous reports that the absence of Dcc produces severe intra-retinal misguidance of RGC axons, and in postnatal brains, that some RGCs still leave the retina and project to visual nuclei with multiple targeting defects. Postnatal loss of RGCs also ensues and the photoreceptor layer does not develop properly, with rosette-like structures forming in the outer retina and reduced photoreceptor function. Both the projection and targeting defects and the photoreceptor results are apparent with a tissue clearing protocol the authors named "EyeDISCO" that provides striking view of the entire visual system and retina in whole mount. The approach overall constitutes a powerful mutant screen for eye and visual pathway perturbations.

**Decision letter after peer review:**

Thank you for submitting your article "Revisiting the role of Dcc in visual system development with a novel eye clearing method" for consideration by *eLife*. Your article has been reviewed by Marianne Bronner as the Senior Editor, a Reviewing Editor, and three reviewers. The following individuals involved in review of your submission have agreed to reveal their identity: Anand Swaroop (Reviewer #2).

The reviewers have discussed the reviews with one another and the Reviewing Editor has drafted this decision to help you prepare a revised submission.

Summary:

The reviewers were overall enthusiastic about your study, in which you generate an eye-specific DCC mutant mouse to more completely characterize the role of DCC in visual system development, especially at postnatal stages. Given that the total knockout dies at birth, you nicely demonstrate that DCC removal not only affects the exit of axons from the eye, as assumed for years, but severe intra-retinal misguidance of RGC axons occurs, unusually, some axons extending within the retina in between the RPE and photoreceptor layers. Further, even with the ensuing optic nerve hypoplasia, a subset of RGCs innervate retinorecipient regions in apparently display normal innervation patterns. The conditional loss of DCC in the eye also results in the novel phenotype of RGC loss and photoreceptor dysplasia in adult mice.

Your paper also includes a welcome optimization of a clearing protocol, EyeDISCO, that allows total removal of RPE pigment by bleaching and therefore a better visualization of the clarified retina and the eye-to-brain. The study thus expands our view of the role of DCC during axon pathway development and maintenance, and a bonus of an important methodology. The paper is a hybrid data presentation and tools and resources, and both aspects will be of high interest to the vision community.

However, even though the reviewers thought your paper suitable for *eLife*, they had numerous comments on the organization of the data and narrative, and encourage you to make the text more concise and better organized to highlight the main points.

They further thought that the clearing protocol, however useful, is too long and disrupts the flow of the paper, making the Results section a bit incoherent. We suggest putting the clearing methods in the Materials and methods section. In addition, the novelty of the new clearing/bleaching methods is diluted by presenting a greater proportion of the data using traditional flatmount/sectioning techniques.

Other comments to be addressed include clarifying the RGC projection phenotype to the MTN, explanations for why only certain RGCs make it to targets, and better resolution on photoreceptor loss, i.e., rods vs cones.

Finally, while the quantitative analyses are much appreciated, presenting the numbers within the Results section is distracting; these data could be presented in the figure legends.

Essential revisions:

We have included the reviews at the end of this email so that you can better amend the text. All of the listed points are deemed essential but are in the main textual.*Reviewer #1:*Vigouroux et al. here generate an eye-specific DCC mutant mouse to more completely characterize the role of DCC in visual system development. First, they analyze DCC expression during early development of the eye. Using their conditional DCC strategy, they find that RGCs lacking DCC display altered axonal trajectories, with a significant percentage extending toward the apical surface of the retina. As previously appreciated, these mice also have severe hypoplasia of the optic nerve. The authors note that not every RGC axon trajectory is altered, as some retinorecipient regions apparently display normal innervation patterns. Novel findings include the observation of severe retinal thinning, RGC loss, and photoreceptor dysplasia in adult mutant mice. In addition, they also present a novel whole eye tissue clearing method which they use to draw several of the conclusions made in this paper.

This study describes in great detail a role for DCC in the developing visual system, providing important data which complements that obtained previously by others using null DCC alleles. Though it was previously known that DCC interactions with netrin-1 are required for a large percentage of RGC axons to exit the eye early in development, it was unclear how DCC functions postnatally and into adulthood. Additionally, the presentation of a novel eye clearing model that allows visualization of axon trajectories in 3D space strengthens the conclusions drawn about DCC function in this paper. This clearing method will certainly aid in future studies of this type and will become a valuable resource for the field. For all of these reasons, this paper is appears to be suitable for publication *eLife*, following attention to points raised below.

1) Please comment as to why one side has a higher innervation density as indicated by CTB labeling in Figure 6B? One would expect that both projection patterns would be equally affected in this genetic background. Is this just animal to animal variability?

2) The point is made that there are apparently differential degrees of retinorecipient innervation defects in the eye-specific DCC mutant, however this seems to be based on the intensity CTB labeled projections (AOS targets (not only the MTN), which are not as densely innervated, appear weaker, as one might expect). The authors should provide more explanation to rule out issues relating to saturation of signaling in more densely innervated targets as opposed to those that receive innervation from many few RGCs. If there are is no additional evidence to support RGC subtype-specific effects resulting from the loss of DCC then this conclusion must be qualified.

3) Please speculate as to why different classes of RGCs and their projection patterns are thought to be differentially affected in these mutants? MTN vs. SC vs. LGN vs. SCN?

4) Please comment on whether, and to what degree, shrinkage is observed of the tissue using this novel clearing approach?

*Reviewer #2:*Vigouroux et al. have re-assessed the function of DCC during retinal development using a new eye-specific Dcc mutant mouse (by Dkk-Cre) to bypass lethality and by introducing a new tissue clearing protocol called "EyeDISCO" to remove melanin. Previously reported Dcc-knockout mice die a few hrs after birth, making it difficult to study DCC function in postnatal stages of visual system development. Using the new Dcc mutant that removes Dcc specifically in retinal progenitors, the authors are able to demonstrate that the absence of DCC results in RGC guidance defects (made possible for EyeDISCO) and postnatal loss of RGCs. The authors also describe the phenotype of kanga/- mutant mouse. Finally, the eye-specific Dcc mutant also shows rosette like structures in the outer retina and reduced rod function.

Overall, the results are significant. The data justify the conclusions. A major focus appears to be on eye-clearing methodology, which seems to be better than another published recently by Hennings et al., 2019, and demonstrating its use in elucidating the mutant phenotype that was previously difficult to observe. The phenotypes being described for the Dcc eye mutant provide new clues about its function. This is an appropriate article for Tools and Resources.

However, the story seemed to lack coherence and finesse. The main points could be better highlighted/organized. The novelty of the new clearing/bleaching methods is diluted by much of the data obtained via tradition flatmount/sectioning techniques.

Overall, the text is too long. The manuscript, especially Results section, requires tightening and can be made concise. While the quantitative analyses are much appreciated, too many numbers within the Results section create distraction. Similarly, too much of background and methodological detail within Results make it incoherent. Some of these could be shifted to Introduction or Methods section. The quantification numbers could be presented in the figure legends.

It's not clear what value the Dcc(kanga) model had for the study. Even if there was more of a difference, a direct comparison with the Dcc-cKO would be more useful than comparison to control only.

The photoreceptor part is quite interesting but not much explored. The rosettes are seen in a few other retinal mutants (such as Nrl-knockout and rd7). The authors should expand more in Discussion section on DCC function in photoreceptors and on how its loss leads to rosettes and loss of rods. The authors suggest that the defect is somewhat limited to s-cones but the authors should stain for m-cones.

It's not clear why cones would be affected more than rods, as it seems based on ERG data. The authors propose in the discussion that it could be a secondary defect resulting from misguided RGCs and potentially defective OLM. Why not co-stain for RGCs or OLM markers with PR markers? This would make a better case since the RGC defect is variable and we can't be sure that there are misguided RGCs in the same regions where the rosettes are. Why PRs are reduced in number? Seems to be developmental defect, not degeneration.

Reviewer #3:

In this work the authors use cutting edge technologies to visualize all the defects produced as a consequence of removing Dcc specifically from the retina. They first confirm previous reports showing that the absence of Dcc produces severe intra-retinal misguidance of RGC axons. Then, they discover that a subset of RGCs are still able to leave the retina and reach the visual nuclei where they project showing multiple targeting defects. They further notice defects in the photoreceptors layer, a phenotype not previously reported in Dcc null mice. To perform many of these analyses they optimize a protocol that allows total removal of pigment and therefore a better visualization of the clarified retina.

There are several reasons that make this work potentially suitable for this journal. First, it reveals for the first time that the absence of retinal Dcc does not only affect the exit of visual axons from the eye, as it has been assumed for years. Second, they conclusively show that the function of Dcc in the retina is restricted to a subpopulation, not to all, RGCs and suggest that Dcc is important for the targeting of retinal axon projections to the medial terminal nucleus in the brain. They also discover that Dcc is expressed in photoreceptors and it is necessary for the proper formation of the photoreceptors layer.

There are however several aspects of the paper that need to be clarified:

1) As the authors point out, the rosettes observed in the ventral retina are likely a consequence of the intra-retinal misrouting of RGC axons. In Figure 5Q, the layer of photoreceptors looks much thinner than in the controls. Is this phenotype homogeneous across the retina or it is also only restricted to the ventral part? In Figure 1—figure supplement 1I is shown that Dcc is expressed in Crx cells (Crx is a cone-rod specific TF) and because the Dkk3-cre line also removes genes from photoreceptors, it is possible that photoreceptors are being intrinsically and homogeneously affected through the entire retina and not only in the ventral retina as a consequence of RGC misrouting.

2) Subsection “Dcc intracellular signaling is required for retinal projection targeting in the Brain”. The authors infer a novel role for Dcc signaling in RGC guidance to the MTN from the projection phenotype to the MTN in the Dcc kanga mutants. Why is this population particularly affected? From what retinal region do the RGC projecting to the MTN come from? It would be possible to perform retrograde labeling from this nucleus to answer this question. If they all come from a particular region perhaps these cells are more affected because they have higher levels of Dcc?

3) It exists also the possibility that the phenotypes observed in the visual nuclei of Dcc mutants are the consequence of having fewer RGCs. In the main visual nuclei it seems likely that rather than a role for Dcc in guidance or targeting, the remaining RGC axon terminals arborize in a different manner because there is less competition between neighboring RGC axons. As the authors state, there is not much information about the timing, guidance or targeting mechanisms of visual axons projecting to the MTN and therefore, this possibility should be at least discussed.

4) I believe is not a good idea to make such a big deal about the optimization of the protocol to remove melanin. What is described in this manuscript is basically an improvement of previous protocols. Therefore, I think that, although is a nice and important advance and it should be mentioned, the part dedicated to explain the protocol is too long and distracts from the main message of the paper. It would be better to make a detailed description of the protocol in a technical or more specialized journal.

[Editors' note: further revisions were suggested prior to acceptance, as described below.]

Thank you for resubmitting your work entitled "Revisiting the role of Dcc in visual system development with a novel eye clearing method" for further consideration by *eLife*. Your revised article has been evaluated by Marianne Bronner (Senior Editor) and a Reviewing Editor.

All three of the reviewers are satisfied with your attempts to clarify the text and figures and with your explanations regarding the points raised in the reviews.

Your manuscript presents important data re-assessing the function of DCC during retinal development using a new eye-specific Dcc mutant mouse (Dkk-Cre) that removes Dcc specifically in retinal progenitors, thereby bypassing lethality at birth. First, you confirm previous reports showing that the absence of Dcc produces severe intra-retinal misguidance of RGC axons, and with the ability to examine postnatal brains, you demonstrate that some RGCs are still able to leave the retina and project to visual nuclei have multiple targeting defects. Second, the absence of DCC results in postnatal loss of RGCs, and because DCC is expressed in photoreceptors, you show that in its absence, the photoreceptor layer does not develop properly, with rosette-like structures forming in the outer retina and reduced photoreceptor function. Importantly, both the projection and targeting defects and the photoreceptor results were made apparent by a tissue clearing protocol you have developed called "EyeDISCO", providing an overview of the entire visual system and retina. Overall the data on the visual projections and photoreceptors are valuable and the presentation properly detailed and quantified.

The manuscript has been improved but there are some remaining issues that need to be addressed before acceptance, as outlined below:

1) Dcc defects: One reviewer had a remaining concern on how you envision that Dcc/Netrin signaling might act intraretinally in a short-range manner. Are you proposing that Netrin1 is secreted by optic disc cells and diffuses all across the retina? This should be more clearly discussed.

2) Materials and methods section, especially EyeDISCO: you improved the manuscript by moving the methods out of the Results section into the Materials and methods section. Nonethless,

a) Reviewer 2 (comment #1) points out, "The novelty of the new clearing/bleaching methods is diluted by much of the data obtained via tradition flatmount/sectioning techniques." We understand that you do not currently have access to a higher resolution objective compatible with DBE and LSFM to allow for precise whole-eye retinal cell counting, and that you are working on this. In the Discussion section, it would be welcome if you could to comment on the current limitations of the technique/microscopy to view retinal cells.

b) From a positive angle, you could highlight the benefits of the clearing approach for future studies, especially for visualizing axon pathways, than you do currently. For example, subsection “EyeDISCO is a powerful technique to study the eye” read like you are promoting EyeDISCO resolution only for analysis of photoreceptor integrity, and do not cite the quite spectacular visualization of axon tracts and targeting. The approach overall to view entire pathways is phenomenal and used together with visualizing eye defects constitutes a powerful screen for mutants; this message should be "heard" more clearly.

c) More recent successes with pigment bleaching in the eye can be found in Iwai -Takekoshi et al., 2016 and 2018.

3) Flow of the findings, and emphasis: In the reviewer consultation session, all three reviewers thought that the flow of the narrative, and the transitions between subsections in the Results section could be improved.

a) To ameliorate this critique, you might consider changing the order of the subsections in the Results section, keeping the first three as is : subsection “Dcc is broadly expressed in the developing retina”, on Dcc expression, subsection “A novel eye-specific Dcc mutant”, on the novel mutant, subsection “Complexity of RGC guidance defects in Dcc KO revealed by a novel eye clearing method” on the novel clearing method, followed by subsection “Retinal projections in the brain, are altered in eye-specific Dcc mutants” on retinal projections in the brain and even move the last subsection “Dcc intracellular signaling is required for retinal projection targeting in the brain” after this section (thus keeping all the projection analysis together). You could end the Results section with subsection “Eye-specific deletion of Dcc alters retinal layer thickness” on retinal layer thickness and then with subsection “Dcc deletion leads to major retinal dysplasia and visual deficits” on Dcc deletion and retinal dysplasia. Admittedly you would have to alter figure numbers and citations, but you wouldn't have to change content of the figures and generally the Results section would read better.

b) In the Discussion section, subsection “EyeDISCO is a powerful technique to study the eye” on EyeDISCO as a technique to study the eye, once you have amended it as to 2) above, should come last.

c) Significance statement – currently: "An innovative tissue clearing protocol to observe the entire visual system sheds light on a novel function of the axon guidance receptor Dcc and its role in retinal development and maintenance". A suggested amendment that incorporates more aspects of the study: A new eye-specific Dcc mutant combined with an improved clearing protocol for the eye and brain (EyeDISCO) reveals the requirement of the axon guidance receptor Dcc for retinal and pathway development and maintenance.

4) English language issues – Please ask a colleague to go over the text, for clarity, usage and grammar; such editing would better showcase. Your excellent study.

a) Singular vs plural – often not correct (Abstract).

b) Tense not consistent: Abstract: "some ganglion cell axons stalled at the optic disc, whereas others perforate….".

c) Terms: Abstract "retinal pigmented epithelium" should be retinal pigment epithelium.

d) RGC as an adjective: subsection "Dcc is essential for RGCs intraretinal…" should be RGC intraretinal.

e) Many sentences are awkward; for instance:

- Subsection “A novel eye-specific Dcc mutant”: "Dcc+ cells were present in Dcc fl/fl retinas" would sound better as: " Dcc+ cells were found in Dcc fl/fl retinas…"

- Subsection “Complexity of RGC guidance defects in Dcc KO revealed by a novel eye clearing method”: the comma should be replaced by a period.

- Subsection “Dcc intracellular signaling is required for retinal projection targeting in the brain”: The sentence "The AOS was also affected in Dcc mice with a significant reduction in of the volume of the MTN was strongly impacted compared to Dcc littermate controls" should be revised. Does not seem to be correct.

---

## [Author Response]

Essential revisions:We have included the reviews at the end of this email so that you can better amend the text. All of the listed points are deemed essential but are in the main textual.Reviewer #1:Vigouroux et al. here generate an eye-specific DCC mutant mouse to more completely characterize the role of DCC in visual system development. First, they analyze DCC expression during early development of the eye. Using their conditional DCC strategy, they find that RGCs lacking DCC display altered axonal trajectories, with a significant percentage extending toward the apical surface of the retina. As previously appreciated, these mice also have severe hypoplasia of the optic nerve. The authors note that not every RGC axon trajectory is altered, as some retinorecipient regions apparently display normal innervation patterns. Novel findings include the observation of severe retinal thinning, RGC loss, and photoreceptor dysplasia in adult mutant mice. In addition, they also present a novel whole eye tissue clearing method which they use to draw several of the conclusions made in this paper.This study describes in great detail a role for DCC in the developing visual system, providing important data which complements that obtained previously by others using null DCC alleles. Though it was previously known that DCC interactions with netrin-1 are required for a large percentage of RGC axons to exit the eye early in development, it was unclear how DCC functions postnatally and into adulthood. Additionally, the presentation of a novel eye clearing model that allows visualization of axon trajectories in 3D space strengthens the conclusions drawn about DCC function in this paper. This clearing method will certainly aid in future studies of this type and will become a valuable resource for the field. For all of these reasons, this paper is appears to be suitable for publication eLife, following attention to relatively minor points raised below.

We thank the reviewer for his/her positive remarks on our study.

1) Please comment as to why one side has a higher innervation density as indicated by CTB labeling in figure 6B? One would expect that both projection patterns would be equally affected in this genetic background. Is this just animal to animal variability?

We agree with the reviewer that the picture in Figure 6B shows a slight asymmetry in the retinal ganglion cell projections. However, this asymmetry is not significant and just reflects eye-to-eye variability in RGC in a given mouse. Indeed, quantifications of optic nerve volumes extracted from CTB-injected mutants does not show a higher variability than in control mice. For example, optic nerve volumes at P15 (SEM ±1.92x10^6^μm^3^ compared to ±1.92x10^6^μm^3^) and at P30 (SEM ±2.48x10^6^μm^3^ compared to ±2.51x10^6^μm^3^).

However, there may be a slight difference between left and right eye projections that we did not take into account in our quantifications.

2) The point is made that there are apparently differential degrees of retinorecipient innervation defects in the eye-specific DCC mutant, however this seems to be based on the intensity CTB labeled projections (AOS targets (not only the MTN), which are not as densely innervated, appear weaker, as one might expect). The authors should provide more explanation to rule out issues relating to saturation of signaling in more densely innervated targets as opposed to those that receive innervation from many few RGCs. If there are is no additional evidence to support RGC subtype-specific effects resulting from the loss of DCC then this conclusion must be qualified.

We agree with the reviewer that there is still a slight possibility that we are missing some visual projections by injecting CTB in the eye. However, CTB tracing is currently the most common method used to study visual projections in mice and we are not aware that it only partially traces visual projections. In optic nerve regeneration experiments, single CTB+ axons could be reconstructed indicating that the tracer is fluorescent enough to label isolated axons (Li et al., 2015). Even if brightness is increased to the maximum, no additional labeling is seen. If staining was incomplete, it would also mean that there is a specific weak staining of AOS axons. However, we have added this possible point of discrepancy to the discussion in the revised text.

We agree with the reviewer that it would be interesting to further tease apart the mechanism responsible for RGC-specific guidance defects in *Dcc* cKO mice. A good strategy would be to cross the *Dcc* conditional mutants to MTN-specific knock in lines (SPIG1:GFP and Hoxd10:GFP). However, we do not currently have access to these lines in the lab and it would probably take a year to get the results.

In an attempt to begin probing at this question, we have purchased 2 commercial antibodies specific for direction-selective RGCs, Hoxd10 and SPIG1. Unfortunately, after testing multiple immunolabeling protocols, none of these antibodies worked in our hands.

3) Please speculate as to why different classes of RGCs and their projection patterns are thought to be differentially affected in these mutants? MTN vs. SC vs. LGN vs. SCN?

We observe that the intensity of Dcc immunolabelling is homogeneous across the entire retina. Thus, we do not believe that specific RGCs may express differential levels of the protein (based on immunostaining). We can only propose at this time that RGC-specific defects may either be due to co-receptors of Dcc which may be differentially expressed in RGCs of the retina (such as UNC5d)(Murcia-Belmonte et al., 2019). Another hypothesis could be that Dcc-specific ligands, in particular Netrin-1 may be differentially expressed in visual targets in the brain and that Dcc/Netrin-1 signaling does not act equally in all RGCs. This has been added to the Discussion section.

4) Please comment on whether, and to what degree, shrinkage is observed of the tissue using this novel clearing approach?

We thank the reviewer for this comment. To address this point, we have quantified the shrinkage of an adult mouse eye following EyeDISCO clearing. We observe a mild shrinkage of approximately 10%. This shrinkage was isotropic across rostro-caudal, dorso-medial, and medio-lateral axes. This was added to the results (Figure 7—figure supplement 1A-D).

Reviewer #2:Vigouroux et al. have re-assessed the function of DCC during retinal development using a new eye-specific Dcc mutant mouse (by Dkk-Cre) to bypass lethality and by introducing a new tissue clearing protocol called "EyeDISCO" to remove melanin. Previously reported Dcc-knockout mice die a few hrs after birth, making it difficult to study DCC function in postnatal stages of visual system development. Using the new Dcc mutant that removes Dcc specifically in retinal progenitors, the authors are able to demonstrate that the absence of DCC results in RGC guidance defects (made possible for EyeDISCO) and postnatal loss of RGCs. The authors also describe the phenotype of kanga/- mutant mouse. Finally, the eye-specific Dcc mutant also shows rosette like structures in the outer retina and reduced rod function.Overall, the results are significant. The data justify the conclusions. A major focus appears to be on eye-clearing methodology, which seems to be better than another published recently by Hennings et al., 2019, and demonstrating its use in elucidating the mutant phenotype that was previously difficult to observe. The phenotypes being described for the Dcc eye mutant provide new clues about its function. This is an appropriate article for Tools and Resources.

We thank the reviewer for his constructive comments on our manuscript. We also agree that we devoted too much space to the eye clearing methods in the original version. The revised manuscript is more focused on the *Dcc* knockout data.

However, the story seemed to lack coherence and finesse. The main points could be better highlighted/organized. The novelty of the new clearing/bleaching methods is diluted by much of the data obtained via tradition flatmount/sectioning techniques.

We agree that many of the quantifications were carried out on flat-mounted retinas. However, we do not currently have access to a well enough resolved objective compatible with DBE and LSFM to allow for precise whole-eye retinal cell counting. We are currently developing such tools to carry out this precise task. This however would not change the validity results of our quantifications performed on flat-mounted retinas.

Overall, the text is too long. The manuscript, especially Results section, requires tightening and can be made concise. While the quantitative analyses are much appreciated, too many numbers within the Results section create distraction. Similarly, too much of background and methodological detail within Results section make it incoherent. Some of these could be shifted to Introduction or Methods section. The quantification numbers could be presented in the figure legends.

To improve the readability of the paper, and as suggested by the reviewer, we have removed all quantifications from the main text and have moved them to figure legends. Furthermore, we have removed the methodological details of the clearing protocol from the Results section. Despite adding new data, the Results section has been shortened (-500 words).

It's not clear what value the Dcc(kanga) model had for the study. Even if there was more of a difference, a direct comparison with the Dcc-cKO would be more useful than comparison to control only.

We understand that we did not convey well enough the importance of the *Dcc Kanga* mutants. In order to better present the value of these mutants we have re-arranged the text.

Moreover, we agree with the reviewer that a direct comparison between our mutants would be more insightful. To that respect, we have added quantifications directly comparing visual nuclei volumes between *Dcc Kanga* mutants and *Dcc* cKO mice (Figure 8—figure supplement 1A-C).

The photoreceptor part is quite interesting but not much explored. The rosettes are seen in a few other retinal mutants (such as Nrl-knockout and rd7). The authors should expand more in Discussion section on DCC function in photoreceptors and on how its loss leads to rosettes and loss of rods. The authors suggest that the defect is somewhat limited to s-cones but the authors should stain for m-cones.

We agree with the reviewer. In order to better characterize the photoreceptors affected in *Dcc* mutant mice, we have carried out cryosections and whole-mount immunostaining for S- and M-cones as well as for Rods. We observe that both S- and M-cones are present in the vicinity and within the rosettes in both *Dcc* cKO (Figure 7—figure supplement 2A-L) and *Dcc Kanga* mice (Figure 8—figure supplement 1N-O). Furthermore, Rhodopsin is also expressed within rosette structures (Figure 7—figure supplement 2M-P).

Lastly, we have expanded on the function of Dcc in photoreceptor development/maintenance in the Discussion section.

It's not clear why cones would be affected more than rods, as it seems based on ERG data. The authors propose in the discussion that it could be a secondary defect resulting from misguided RGCs and potentially defective OLM. Why not co-stain for RGCs or OLM markers with PR markers? This would make a better case since the RGC defect is variable and we can't be sure that there are misguided RGCs in the same regions where the rosettes are. Why PRs are reduced in number? Seems to be developmental defect, not degeneration.

We agree with the reviewer. In order to emphasize the developmental defect of axon tracts separating the photoreceptors from the retinal pigment epithelium, we have carried out cryosections and labeled for RGC tracts (ßIII tubulin) and photoreceptors (Opn1sw)(Figure 4—figure supplement 1A-D).

We do not believe cones would be more affected than rods in our mutants. To target this, we have carried out immuno-labelling for cones (S and M) as well as rods (Figure 7—figure supplement 2A-P). All photoreceptors are impacted similarly. However, since the proportion of cones to rods in the murine retina is lower, it could explain why we have a stronger defect on the ERG data.

Reviewer #3:In this work the authors use cutting edge technologies to visualize all the defects produced as a consequence of removing Dcc specifically from the retina. They first confirm previous reports showing that the absence of Dcc produces severe intra-retinal misguidance of RGC axons. Then, they discover that a subset of RGCs are still able to leave the retina and reach the visual nuclei where they project showing multiple targeting defects. They further notice defects in the photoreceptors layer, a phenotype not previously reported in Dcc null mice. To perform many of these analyses they optimize a protocol that allows total removal of pigment and therefore a better visualization of the clarified retina.There are several reasons that make this work potentially suitable for this journal. First, it reveals for the first time that the absence of retinal Dcc does not only affect the exit of visual axons from the eye, as it has been assumed for years. Second, they conclusively show that the function of Dcc in the retina is restricted to a subpopulation, not to all, RGCs and suggest that Dcc is important for the targeting of retinal axon projections to the medial terminal nucleus in the brain. They also discover that Dcc is expressed in photoreceptors and it is necessary for the proper formation of the photoreceptors layer.There are however several aspects of the paper that need to be clarified:1) As the authors point out, the rosettes observed in the ventral retina are likely a consequence of the intra-retinal misrouting of RGC axons. In Figure 5Q, the layer of photoreceptors looks much thinner than in the controls. Is this phenotype homogeneous across the retina or it is also only restricted to the ventral part?

We observe that the retinal thinning is homogeneous across the retina. This information was added to the Results section.

In Figure 1—figure supplement 1I is shown that Dcc is expressed in Crx cells (Crx is a cone-rod specific TF) and because the Dkk3-cre line also removes genes from photoreceptors, it is possible that photoreceptors are being intrinsically and homogeneously affected through the entire retina and not only in the ventral retina as a consequence of RGC misrouting.

We thank the reviewer for this valuable question. Indeed, the formation of rosettes is restricted to specific regions in the retina. However, this does not mean that photoreceptor function outside of rosettes is normal. Several observations allude to the possibility that Dcc may play a role in photoreceptor maintenance retina-wide. Firstly, Dcc is transiently expressed in Crx^+^ cells. Furthermore, our ERG data show that the overall physiological response of the retina is altered. Therefore, it is possible that loss of Dcc may impact photoreceptor function outside of rosettes. Lastly, we have carried out quantifications in our *Dcc* cKO mice of the ONL thickness from regions of the retina outside of rosettes. Indeed, we visualize a dramatic thinning of the ONL (Figure 7—figure supplement 2Q-U). However, the observation that rosette formations do not progress overtime may point to the opposite hypothesis that loss of Dcc does not induce degeneration of photoreceptors. These points were added to the Discussion section.

2) Subsection “Dcc intracellular signaling is required for retinal projection targeting in the Brain”. The authors infer a novel role for Dcc signaling in RGC guidance to the MTN from the projection phenotype to the MTN in the Dcc kanga mutants. Why is this population particularly affected? From what retinal region do the RGC projecting to the MTN come from?

Indeed, we observe in the Dcc cKO and Dcc Kanga mice a specific defect in the MTN. RGCs projecting to the MTN are distributed homogeneously across the retina. However, several markers have been identified to specifically label these RGCs. Of note, SPIG1 and Hoxd10 label these cells. We have acquired these antibodies and have unfortunately not been able reveal a specific labelling. Both SPIG1:GFP and Hoxd10:GFP knock-in mouse lines exist. However, they are not available in the lab and would take us a long time to obtain. We have therefore included these points as perspectives in the Discussion section.

It would be possible to perform retrograde labeling from this nucleus to answer this question. If they all come from a particular region perhaps these cells are more affected because they have higher levels of Dcc?

We thank the reviewer for this point. However, on whole-mount retinas, we observed a homogeneous expression of Dcc retina-wide. Furthermore, RGCs projecting to the MTN are spread across the retina, therefore simply carrying out a flat-mount or EyeDISCO of Dcc cKO retinas would not help us isolate this population. Indeed, retro-labelling the MTN with a tracer would allow us to track MTN-specific RGCs. However, whilst already quite technically challenging in wild type mice, carrying out these injections in the Dcc cKO mice would be very difficult since the volume of RGC projections to this nucleus is dramatically reduced.

3) It exists also the possibility that the phenotypes observed in the visual nuclei of Dcc mutants are the consequence of having fewer RGCs. In the main visual nuclei it seems likely that rather than a role for Dcc in guidance or targeting, the remaining RGC axon terminals arborize in a different manner because there is less competition between neighboring RGC axons. As the authors state, there is not much information about the timing, guidance or targeting mechanisms of visual axons projecting to the MTN and therefore, this possibility should be at least discussed.

We agree with the reviewer and have added this point to the Discussion section.

4) I believe is not a good idea to make such a big deal about the optimization of the protocol to remove melanin. What is described in this manuscript is basically an improvement of previous protocols. Therefore, I think that, although is a nice and important advance and it should be mentioned, the part dedicated to explain the protocol is too long and distracts from the main message of the paper. It would be better to make a detailed description of the protocol in a technical or more specialized journal.

We thank the reviewer for this point and suggestion. We have therefore removed all the unnecessary details of the EyeDISCO protocol.

[Editors' note: further revisions were suggested prior to acceptance, as described below.]

The manuscript has been improved but there are some remaining issues that need to be addressed before acceptance, as outlined below:1) Dcc defects: One reviewer had a remaining concern on how you envision that Dcc/Netrin signaling might act intraretinally in a short-range manner. Are you proposing that Netrin1 is secreted by optic disc cells and diffuses all across the retina? This should be more clearly discussed.

We agree with the reviewer that the Dcc/Netrin-1 signaling should be more clearly discussed and therefore modified the Discussion section accordingly.

2) Materials and methods section, especially EyeDISCO: you improved the manuscript by moving the methods out of the Results section into the Materials and methods section. Nonethless,a) Reviewer 2 (comment #1) points out, "The novelty of the new clearing/bleaching methods is diluted by much of the data obtained via tradition flatmount/sectioning techniques." We understand that you do not currently have access to a higher resolution objective compatible with DBE and LSFM to allow for precise whole-eye retinal cell counting, and that you are working on this. In the Discussion section, it would be welcome if you could to comment on the current limitations of the technique/microscopy to view retinal cells.

We thank the reviewer for his constructive remark and added comment on this topic in the Discussion section.

b) From a positive angle, you could highlight the benefits of the clearing approach for future studies, especially for visualizing axon pathways, than. you do currently. For example, subsection “EyeDISCO is a powerful technique to study the eye” read like you are promoting EyeDISCO resolution only for analysis of photoreceptor integrity, and do not cite the quite spectacular visualization of axon tracts and targeting. The approach overall to view entire pathways is phenomenal and used together with visualizing eye defects constitutes a powerful screen for mutants; this message should be "heard" more clearly.

We thank the reviewer for his positive comment and changed the Discussion section correspondingly.

c) More recent successes with pigment bleaching in the eye can be found in Iwai -Takekoshi et al., 2016 and 2018.

We added this method in the Discussion section and explained why we have not considered it.

3) Flow of the findings, and emphasis: In the reviewer consultation session, all three reviewers thought that the flow of the narrative, and the transitions between subsections in the Results section could be improved.a) To ameliorate this critique, you might consider changing the order of the subsections in the Results section, keeping the first three as is : subsection “Dcc is broadly expressed in the developing retina”, on Dcc expression, subsection “A novel eye-specific Dcc mutant”, on the novel mutant, subsection “Complexity of RGC guidance defects in Dcc KO revealed by a novel eye clearing method” on the novel clearing method, followed by subsection “Retinal projections in the brain, are altered in eye-specific Dcc mutants” on retinal projections in the brain and even move the last subsection “Dcc intracellular signaling is required for retinal projection targeting in the brain” after this section (thus keeping all the projection analysis together). You could end the Results section with subsection “Eye-specific deletion of Dcc alters retinal layer thickness” on retinal layer thickness and then with subsection “Dcc deletion leads to major retinal dysplasia and visual deficits” on Dcc deletion and retinal dysplasia. Admittedly you would have to alter figure numbers and citations, but you wouldn't have to change content of the figures and generally the Results section would read better.

We thank the reviewers for this suggestion to improve the flow of the narrative. We changed the Results section accordingly. Nevertheless, these modifications introduced the retinal dysplasia phenotype in the Dcc Kanga mutants before the Dcc cKO. Therefore, we had to modify the figures in order to restore the description of the rosettes in the DKK3creDCC^lox/lox^ mice before the Dcc Kanga mutants.

Figure 6 is now Figure 5

Figure 8A-M is now Figure 6

Figure 8N-Q and Figure 8—figure supplement 1L-O are now combined in Figure 8—figure supplement 4

Figure 8—figure supplement 1 A-E are now Figure6—figure supplement 1

Figure 8—figure supplement 1 F-K are now Figure7—figure supplement 3

Figure 5 is now Figure 7

Figure 5—figure supplement 1 is now Figure 7—figure supplement 1

Figure 5—figure supplement 2 is now Figure 7—figure supplement 2

Figure 7 is now Figure 8

Figure 7—figure supplement 1 is now Figure 8—figure supplement 1

Figure 7—figure supplement 2 is now Figure 8—figure supplement 2

Figure 7—figure supplement 3 is now Figure 8—figure supplement 3

b) In the Discussion section, subsection “EyeDISCO is a powerful technique to study the eye” on EyeDISCO as a technique to study the eye, once you have amended it as to 2. above, should come last.

We have moved the referred part of the Discussion section to the end.

c) Significance statement – currently: "An innovative tissue clearing protocol to observe the entire visual system sheds light on a novel function of the axon guidance receptor Dcc and its role in retinal development and maintenance". A suggested amendment that incorporates more aspects of the study: A new eye-specific Dcc mutant combined with an improved clearing protocol for the eye and brain (EyeDISCO) reveals the requirement of the axon guidance receptor Dcc for retinal and pathway development and maintenance.

We thank the reviewer for this valuable suggestion and changed the significance statement. Yet we had to slightly modify what was suggested as it was exceeding the word limit.

4) English language issues – Please ask a colleague to go over the text, for clarity, usage and grammar; such editing would better showcase. Your excellent study.

We apologized for the English language issues. The text was edited by an English native speaker colleague for improvement.

a) Singular vs plural – often not correct (Abstract).

This was modified in the text.

b) Tense not consistent: Abstract: "some ganglion cell axons stalled at the optic disc, whereas others perforate….".

This was modified in the text.

c) Terms: Abstract "retinal pigmented epithelium" should be retinal pigment epithelium.

This was modified in the text.

d) RGC as an adjective: subsection "Dcc is essential for RGCs intraretinal…" should be RGC intraretinal.e) Many sentences are awkward; for instance:- Subsection “A novel eye-specific Dcc mutant”: "Dcc+ cells were present in Dcc fl/fl retinas" would sound better as: " Dcc+ cells were found in Dcc fl/fl retinas…"

This was modified in the text.

- Subsection “Complexity of RGC guidance defects in Dcc KO revealed by a novel eye clearing method”: the comma should be replaced by a period.

This mistake was modified in the text.

- Subsection “Dcc intracellular signaling is required for retinal projection targeting in the brain”: The sentence "The AOS was also affected in Dcc mice with a significant reduction in of the volume of the MTN was strongly impacted compared to Dcc littermate controls" should be revised. Does not seem to be correct.

This was corrected in the text.